# One Explanation is Not Enough: Structured Attention Graphs for Image Classification

**Vivswan Shitole, Li Fuxin, Minsuk Kahng, Prasad Tadepalli, Alan Fern**
Oregon State University
`{shitolev, lif, minsuk.kahng, tadepall, alan.fern}@oregonstate.edu`

## Abstract

Saliency maps are popular tools for explaining the decisions of convolutional neural networks (CNNs) for image classification. Typically, for each image of interest, a single saliency map is produced, which assigns weights to pixels based on their importance to the classification. We argue that a single saliency map provides an incomplete understanding since there are often many other maps that can explain a classification equally well. In this paper, we propose to utilize a beam search algorithm to systematically search for multiple explanations for each image. Results show that there are indeed multiple relatively localized explanations for many images. However, naively showing multiple explanations to users can be overwhelming and does not reveal their common and distinct structures. We introduce *structured attention graphs* (SAGs), which compactly represent sets of attention maps for an image by visualizing how different combinations of image regions impact the confidence of a classifier. An approach to computing a compact and representative SAG for visualization is proposed via diverse sampling. We conduct a user study comparing the use of SAGs to traditional saliency maps for answering comparative counterfactual questions about image classifications. Our results show that user accuracy is increased significantly when presented with SAGs compared to standard saliency map baselines.

## 1 Introduction

With the emergence of convolutional neural networks (CNNs) as the most successful learning paradigm for image classification, the need for human understandable explanations of their decisions has gained prominence. Explanations lead to a deeper user understanding and trust of the neural network models, which is crucial for their deployment in safety-critical applications. They can also help identify potential causes of misclassification. An important goal of explanation is for the users to gain a *mental model* of the CNNs, so that the users can understand and predict the behavior of the classifier[17] in cases that have not been seen. A better mental model would lead to appropriate trust and better safeguards of the deep networks in the deployment process.

A popular line of research towards this goal has been to display attention maps, sometimes called saliency maps or heatmaps. Most approaches assign weights to image regions based on the importance of that region to the classification decision, which is then visualized to the user. This approach implicitly assumes that a *single saliency map* with region-specific weights is sufficient for the human to construct a reasonable mental model of the classification decision for the particular image.

We argue that this is not always the case. Fig. 1(d-f) show three localized attention maps highlighting different regions. Each of these images, if given as input to the CNN, results in a very confident prediction of the correct category. However, this information is not apparent from a single saliency map as produced by current methods (Fig. 1(b-c)). This raises several questions: How many images have small localized explanations (i.e., attention maps) that lead to high confidence predictions? Are

35th Conference on Neural Information Processing Systems (NeurIPS 2021).

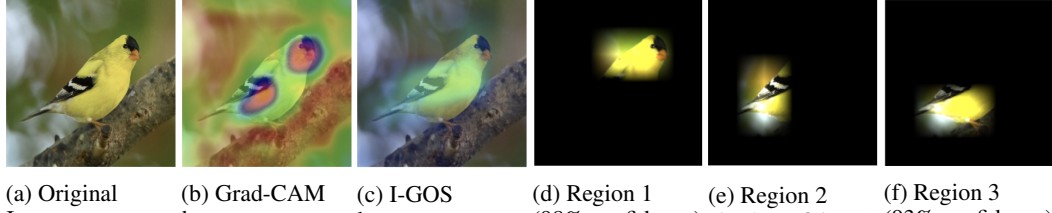

| (a) Original Image | (b) Grad-CAM heatmap | (c) I-GOS heatmap | (d) Region 1 (98% confidence) | (e) Region 2 (97% confidence) | (f) Region 3 (93% confidence) |

Figure 1: An image (a) predicted as Goldfinch with two saliency maps (b) and (c) obtained from different approaches as explanations for the classifier's (VGGNet [27]) prediction. Each of these saliency maps creates a narrow understanding of the classifier. In (d), (e) and (f), we present three diverse regions of the image that might not be deemed important by the singleton saliency maps (b) and (c), and yet are classified as the target class with high confidence by the same classifier

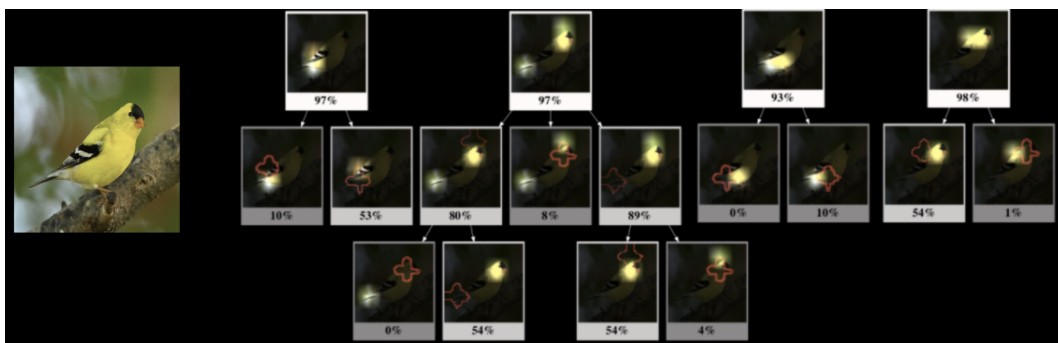

Figure 2: Example of a SAG. For the goldfinch image on the left, a SAG on the right is structured as a directed acyclic graph with each root node representing a minimal region of the image sufficient to achieve a high confidence for the classifier's prediction. Each child node is obtained by deleting a patch (denoted by red contour) from the parent, causing a drop in the classifier's confidence. A significant drop in confidence implies the removed patch was of high importance to the classifier. More examples of SAGs are provided in the appendix

there multiple distinct high confidence explanations for each image, and if so, how to find them? How can we efficiently visualize multiple explanations to users to yield deeper insights?

The first goal of this paper is to systematically evaluate the sizes and numbers of high-confidence local attention maps of CNN image classifications. For this purpose, rather than adopting commonly used gradient-based optimization approaches, we employ discrete search algorithms to find multiple high-confidence attention maps that are distinct in their coverage.

The existence of multiple attention maps shows that CNN decisions may be more comprehensively explained with a logical structure in the form of disjunctions of conjunctions of features represented by local regions instead of a singleton saliency map. However, a significant challenge in utilizing this as an explanation is to come up with a proper visualization to help users gain a more comprehensive mental model of the CNN. This leads us to our second contribution of the paper, *Structured Attention Graphs (SAGs)* [1] , which are directed acyclic graphs over attention maps of different image regions. The maps are connected based on containment relationships between the regions, and each map is accompanied with the prediction confidence of the classification based on the map (see Fig. 2 for an example). We propose a diverse sampling approach to select a compact and diverse set of maps for SAG construction and visualization.

This new SAG visualization allows users to efficiently view information from a diverse set of maps, which serves as a novel type of explanation for CNN decisions. In particular, SAGs provide insight by decomposing local maps into sub-regions and making the common and distinct structures across maps explicit. For example, observing that the removal of a particular patch leads to a huge drop in the confidence suggests that the patch might be important in that context.

---

[1]Source code for generating SAGs: https://github.com/viv92/structured-attention-graphs

Our visualization can also be viewed as representing a (probabilistic) Monotone Disjunctive Normal Form (MDNF) Boolean expression, where propositional symbols correspond to primitive image regions we call 'patches'. Each MDNF expression is a disjunction of conjunctions, where any one of the conjunctions (e.g., one of the regions in Fig. 1) is sufficient for a high confident classification. Following [13], we call these minimal sufficient explanations (MSEs). Each conjunction is true only when all the patches that correspond to its symbols are present in the image.

We conducted a large-scale user study (100 participants total) to compare SAGs to two saliency map methods. We wondered if participants can answer challenging counterfactual questions with the help of explanations , e.g., how a CNN model classifies an image *if* parts of the image are occluded . In our user study, participants were provided two different occluded versions of the image (i.e., different parts of the image are occluded ) and asked to choose one that they think would be classified more positively. Results show that when presented with SAG, participants correctly answer significantly more of these questions compared to the baselines, which suggests that SAGs help them build better mental models of the behavior of the classifier on different subimages.

In summary, our contributions are as follows:

- With a beam search algorithm, we conducted a systematic study of the sizes and numbers of attention maps that yield high confidence classifications of a CNN (VGGNet [27]) on ImageNet [7]. We showed that the proposed beam search algorithm significantly outperforms Grad-CAM and I-GOS in its capability to locate small attention maps to explain CNN decisions.

- We introduce Structured Attention Graphs (SAGs) as a novel representation to visualize image classifications by convolutional neural networks.

- We conducted a user study demonstrating the effectiveness of SAGs in helping users gain a deeper understanding of CNN's decision making.

## 2   Related Work

Much recent work on interpretability of CNNs is based on different ways to generate saliency maps depicting the importance of different regions to the classification decisions. These include *gradient-based methods* that compute the gradient of the outputs of different units with respect to pixel inputs [31, 26, 28, 25, 29, 2, 25, 32, 24], *perturbation-based methods*, which perturb parts of the input to see which ones are most important to preserve the final decision [5, 9], and *concept-based methods*, which analyze the alignment between individual hidden neurons and a set of semantic concepts [3, 14, 33]. Importantly, they all generate a single saliency map for the image and have been found to be brittle and unreliable [15, 10].

Another popular approach is LIME [21], which constructs simplified interpretable local classifiers consistent with the black-box classifier in the neighborhood of a single example. However, the local classifier learns a single linear function, which is sufficient to correctly classify the image but does not guarantee consistency with the classifier on its sub-images. More recently, Anchors [22] learns multiple if-then-rules that represent sufficient conditions for classifications. However, this work did not emphasize image classification and did not systematically study the prevalence of multiple explanations for the decisions of CNNs. The if-then-rules in Anchors can be thought of as represented by the root nodes in our SAG. SAGs differ from them by sampling a diverse set for visualization, as well as by additionally representing the relationships between different subregions in the image and their impact on the classification scores of the CNN. The ablation study of Section 5.3 shows that SAGs enable users to better understand the importance of different patches on the classification compared to Anchors-like rules represented by their root nodes.

Some prior work identifies explanations in terms of minimal necessary features [8] and minimal sufficient features [5]. Other work generates counterfactuals that are coherent with the underlying data distribution and provides feasible paths to the target counterfactual class based on density weighted metrics [19]. In contrast, our work yields multiple explanations in terms of minimal sufficient features and visualizes the score changes when some features are absent – simultaneously answering multiple counterfactual questions.

Network distillation methods that compile a neural network into a boolean circuit [4] or a decision tree [16] often yield uninterpretable structures due to their size or complexity. Our work balances the

information gain from explanations with the interpretability of explanations by providing a small set of diverse explanations structured as a graph over attention maps.

# 3 Investigating Image Explanations

In this section, we provide a comprehensive study of the number of different explanations of the images as well as their sizes. As the number of explanations might be combinatorial, we limit the search space by subdividing each image into $49 = 7 \times 7$ patches, which corresponds to the resolution utilized in Grad-CAM [24]. Instead of using a heatmap algorithm, we propose to utilize search algorithms to check the CNN (VGGNet [27]) predictions on many combinations of patches in order to determine whether they are able to explain the prediction of the CNN by being a minimum sufficient explanation, defined as having a high prediction confidence from a minimal combination of patches w.r.t. using the full image. The rationale is that if the CNN is capable of achieving the same confidence from a subimage, then the rest of the image may not add substantially to the classification decision. This corresponds to common metrics used in evaluating explanations [23, 18, 20], which usually score saliency maps based on whether they could use a small highlighted part of the image to achieve similar classification accuracy as using the full image. This experiment allows us to examine multiple interesting aspects, such as the minimal number of patches needed to explain each image, as well as the number of diverse explanations by exploring different combinations of patches. The ImageNet validation dataset of $50,000$ images is used for our analysis.

Formally, we assume a black-box classifier $f$ that maps $X \to [0, 1]^C$, where $X$ is an instance space and $C$ is a set of classes. If $x \in X$ is an instance, we use $f_c(x)$ to denote the output class-conditional probability on class $c \in C$. The predicted class-conditional probability is referred as *confidence* of the classification in the rest of the paper. In this paper we assume $X$ is a set of images. Each image $x \in X$ can be seen as a set of pixels and is divided into $r^2$ non-overlapping primitive regions $p_i$ called 'patches,' i.e., $x = \cup_{i=1}^{r^2} p_i$, where $p_i \cap p_j = \emptyset$ if $i \neq j$. For any image $x \in X$, we let $f^*(x) = \text{argmax}_c f_c(x)$ and call $f^*(x)$ the target class of $x$. We associate the part of the image in each patch with a propositional symbol or a *literal*. A *conjunction* $N$ of a set of literals is the image region that corresponds to their union. The *confidence* of a conjunction is the output of the classifier $f$ applied to it, denoted by $f_c(N)$. We determine this by running the classifier on a perturbed image where the pixels in $x \setminus N$ are either set to zeros or to a highly blurred version of the original image. The latter method is widely used in saliency map visualization methods to remove information without creating additional spurious boundaries that can distort the classifier predictions [9, 18, 20].

A minimal sufficient explanation (MSE) of an image $x$ as class $c$ w.r.t. $f$ is defined as a minimal conjunction/region that achieves a high prediction confidence ($f_c(N_i) > P_h f_c(x)$) w.r.t. using the entire image, where we set $P_h = 0.9$ as a- sufficiently high fraction in our experiments. That is, if we provide the classifier with only the region represented by the MSE , it will yield a confidence at that is at least $90\%$ of the confidence for the original (unoccluded) image $x$ as input. Often we will be most interested in MSEs for $c = f^*(x)$.

## 3.1 Finding MSEs via Search

A central claim of the paper we purport to prove is that the MSEs are not unique, and can be found by systematic search in the space of subregions of the image. The search objective is to find the minimal sufficient explanations $N_i$ that score higher than a threshold where no proper sub-regions exceed the threshold, i.e., find all $N_i$ such that:

$$f_c(N_i) \geq P_h f_c(x), \max_{n_j \subset N_i} f_c(n_j) < P_h f_c(x) \tag{1}$$

for some high probability threshold $P_h$.

But such a combinatorial search is too expensive to be feasible if we treat each image pixel as a patch. Hence we divide the image into a coarser set of non-overlapping patches. One could utilize a superpixel tessellation of an image to form the set of coarser patches. We adopt a simpler approach: we downsample the image into a low resolution $r \times r$ image. Each pixel in the downsampled image corresponds to a coarser patch in the original image. Hence a search on the downsampled image is computationally less expensive. We set the hyperparameter $r = 7$ in all our experiments. Further, to use an attention map $M$ as a heuristic for search on the downsampled image, we perform average

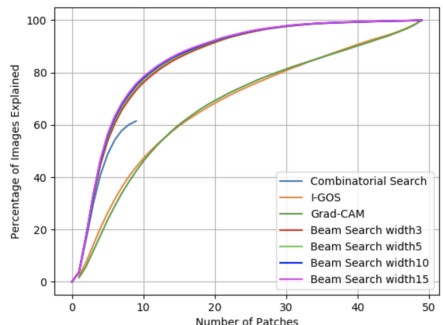

Figure 3: Percentage of images explained by different number of patches.

Table 1: Number of diverse MSEs obtained by allowing for different degrees of overlap.

| | Overlap = 0 | | |
|---|---|---|---|
| Method | Mean | Variance | Mode |
| CombS | 1.56 | 0.69 | 1 |
| BeamS-3 | 1.72 | 0.98 | 1 |
| BeamS-15 | 1.87 | 1.01 | 2 |
| | Overlap = 1 | | |
| Method | Mean | Variance | Mode |
| CombS | 3.16 | 5.82 | 1 |
| BeamS-3 | 4.18 | 7.24 | 2 |
| BeamS-15 | 4.51 | 7.22 | 3 |

pooling on $M$ w.r.t. each patch $p_j$. This gives us an attention value $M(p_j)$ for each patch, hence constituting a coarser attention map. Once the attention map is generated in low resolution, we use bilinear upsampling to upsample it to the original image resolution to be used as a mask. Bilinear upsampling creates a slightly rounded region for each patch which avoids sharp corners that could be erroneously picked up by CNNs as features.

We analyze two different search methods for finding the MSEs:

**Restricted Combinatorial Search:** Combinatorial search constrains the size of the MSE to $k$ patches and finds the MSEs $N_k$ by searching for all combinations (conjunctions) of $k$ patches that satisfy the criterion in Equation 1. However, such a combinatorial search over the entire downsampled image will be of the order $\binom{r^2}{k}$, which is computationally expensive. Hence, we first prune the search space by selecting the $m$ most relevant patches, where the relevance of each patch $p_j$ is given by an attention map as $M(p_j)$, and then carry out a combinatorial search. We set $m = 10$ and vary $0 < k < m$ as hyperparameters. These hyperparameter choices allow the combinatorial search to complete in reasonable time.

**Beam Search:** Beam search searches for a set of at most $w$ MSEs $S = \{N_1, N_2, ..., N_w\}$ by maintaining a set of $w$ distinct conjunctions of patches $S^i = \{N_1^i, N_2^i, ..., N_w^i\}$ as states at the $ith$ iteration. It adds a patch to each conjunction to obtain a new set of $w$ distinct conjunctions $S^{i+1} = \{N_1^{i+1}, N_2^{i+1}, ..., N_w^{i+1}\}$ as successor states for the next iteration, until they satisfy the criterion in equation 1 to yield the set $S$. This is similar to the traditional beam search with beam width $w$, but we leverage the attention map $M$ for generating the successor states. More concretely, the search is initialized by selecting the highest weighted $w$ patches from the attention map as the set of initial $w$ states $S^0 = \{N_1^0, N_2^0, ..., N_w^0\}$. At any iteration $i$, for each state $N_j^i \in S^i$, we generate $q$ candidate successor states $\{Q_{j1}^i, Q_{j2}^i, ..., Q_{jq}^i\}$ by adding the $q$ highest weighted patches in the attention map that are not already in $N_j^i$. By doing this for each of the $w$ states in $S^i$, we generate a set of $w \times q$ candidate successor states. We obtain the classification score for each candidate successor state $f_c(Q_{jx}^i)$ and select the highest scoring $w$ states as the successor states $S^{i+1} = \{N_1^{i+1}, N_2^{i+1}, ..., N_w^{i+1}\}$. We chose $q = 15$ as a hyperparameter. This choice of value for the hyperparameter allows the beam search to complete in reasonable time.

### 3.2 Size of Minimal Sufficient Explanations

Each search method yields a set of MSEs constituting multiple minimal regions of an image sufficient for the black-box classifier to correctly classify the image with a high confidence. We measure the size of these minimal regions in terms of the number of patches they are composed of.

Fig. 3 shows these plots for different search methods on the VGG network. Results on ResNet-50 are shown in the appendix. For each chosen size $k$, we plot the cumulative number of images whose MSE has a size $\leq k$. We see that 80% images of the ImageNet validation dataset have at least one MSE comprising of 10 or less patches. This implies that 80% images of the dataset can be confidently classified by the CNN using a region of the image comprising of just 20% of the area of the original image, showing that in most cases CNNs are able to make decisions based on local information instead of looking at the entire image. The remaining 20% of the images in the dataset have MSEs that

| | Overlap = 0 | | | | | Overlap = 1 | | | | |
|---|---|---|---|---|---|---|---|---|---|---|
| Method ⟍ # of diverse MSEs | 1 | 2 | 3 | 4 | 5 | 1 | 2 | 3 | 4 | 5 |
| CombS | 87% | 9% | 2% | 0% | 0% | 79% | 5% | 4% | 2% | 2% |
| BeamS-3 | 84% | 11% | 2% | 0% | 0% | 65% | 15% | 6% | 3% | 2% |
| BeamS-15 | 79% | 15% | 2% | 1% | 0% | 59% | 16% | 8% | 5% | 2% |

Table 2: Percentage of images versus number of diverse MSEs obtained by allowing for different degrees of overlap.

fall in the range of 11-49 patches (20% - 100% of the original image). Besides, one can see that many more images can be explained via the beam search approach w.r.t. conventional heatmap generation approaches, because the search algorithm evaluated combinations more comprehensively than these heatmap approaches and is less likely to include irrelevant regions. For example, at 10 patches, beam search with all beam sizes can explain about $80\%$ of ImageNet images, whereas Grad-CAM and I-GOS can only explain about $50\%$. Although beam search as an saliency map method is limited to a low resolution whereas some other saliency map algorithms can generate heatmaps at a higher resolution, this result shows that the beam search algorithm is more effective than traditional saliency map approaches at a low resolution.

### 3.3 Number of Diverse MSEs

Given the set of MSEs obtained via different search methods, we also analyze the number of diverse MSEs that exist for an image. Two MSEs of the same image are considered to be diverse if they have less than two patches in common. Table 1 provides the statistics on the number of diverse MSEs obtained by allowing for different degrees of overlap across the employed search methods. We see that images tend to have multiple MSEs sufficient for confident classification, with $\approx 2$ explanations per image if we do not allow any overlap, and $\approx 5$ explanations per image if we allow a 1-patch overlap. Table 2 provides the percentage of images having a particular number of diverse MSEs. This result confirms our hypothesis that in many images CNNs have more than one way to classify each single image. In those cases, explanations based on a single saliency map pose an incomplete picture of the decision-making of the CNN classifier.

## 4 Structured Attention Graphs

From the previous section, we learned about the prevalence of multiple explanations. How can we then, effectively present them to human users so that they can better build mental models of the behavior of image classifiers?

This section introduces *structured attention graphs (SAGs)*, a new way to compactly represent sets of attention maps for an image by visualizing how different combinations of image regions impact the confidence of a classifier. Fig. 2 shows an example. A SAG is a directed acyclic graph whose nodes correspond to sets of image patches and edges represent *subset* relationships between sets defined by the removal of a single patch. The root nodes of SAG correspond to sets of patches that represent *minimal sufficient explanations* (MSEs) as defined in the previous section. Typically, the score of the root node $N_i$ is higher than all its children $n_j \subset N_i$. The size of the drop in the score may correspond to the importance of the removed patch $N_i \setminus n_j$. Under the reasonable assumption that the function $f$ is monotonic with the set of pixels covered by the region, the explanation problem generalizes learning Monotone DNF (MDNF) boolean expressions from membership (yes/no) queries, where each disjunction corresponds to a root node of the SAG, which in turn represents a conjunction of primitive patches. Information-theoretic bounds imply that the general class of MDNF expressions is not learnable with polynomial number of membership queries although some special cases are learnable [1]. The next two subsections describe how a SAG is constructed.

### 4.1 Finding Diverse MSEs

We first find multiple candidate MSEs $\tilde{N}_{candidates} = \{\tilde{N}_1, ..., \tilde{N}_t\}$, for some $t > 1$ through search. We observe that the obtained set $\tilde{N}_{candidates}$ often has a large number of similar MSEs that share

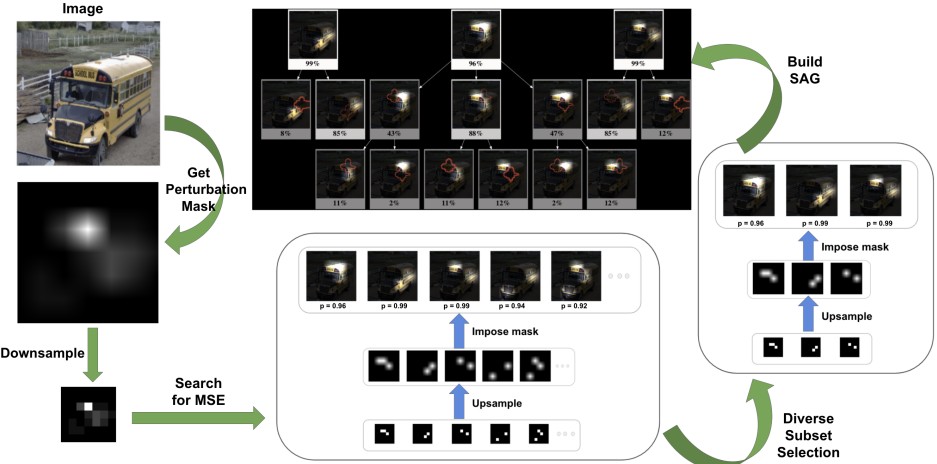

Figure 4: Illustration of the steps for generating a SAG (on top middle) from a given image (on top left).

a number of literals. To minimize the cognitive burden on the user and efficiently communicate relevant information with a small number of MSEs, we heuristically prune the above set to select a small diverse subset. Note that we prefer a diverse subset (based on dispersion metrics) over a representative subset (based on coverage metrics). This choice was based on the observation that even a redundant subset of candidates $\tilde{N}_{\text{redundant}} \subset \tilde{N}_{\text{candidates}}$ can achieve high coverage when the exhaustive set $\tilde{N}_{\text{candidates}}$ has high redundancy. But $\tilde{N}_{\text{redundant}}$ has lower information compared to a diverse subset of candidates $\tilde{N}_{\text{diverse}} \subset \tilde{N}_{\text{candidates}}$ obtained by optimizing a dispersion metric.

More concretely, we want to find an information-rich diverse solution set $\tilde{N}_{\text{diverse}} \subset \tilde{N}_{\text{candidates}}$ of a desired size $c$ such that $|\tilde{N}_i \cap \tilde{N}_j|$ is minimized   for all $\tilde{N}_i, \tilde{N}_j \in \tilde{N}_{\text{diverse}}$ where $i \neq j$. We note that $\tilde{N}_{\text{diverse}}$ can be obtained by solving the following subset selection problem:

$$\tilde{N}_{\text{diverse}} = \underset{X \subseteq \tilde{N}_{\text{candidates}}, |X|=c}{\text{argmin}} \psi(X),$$
$$where \quad \psi(X) = \max_{\tilde{N}_i, \tilde{N}_{j \neq i} \in X} |\tilde{N}_i \cap \tilde{N}_j|$$

For any subset $X$ of the candidate set, $\psi(X)$ is the cardinality of the largest pairwise intersection over all member sets of $X$. $\tilde{N}_{\text{diverse}}$ is the subset with minimum value for $\psi(X)$ among all the subsets $X$ of a fixed cardinality $c$. Minimizing $\psi(X)$ is equivalent to maximizing a dispersion function, for which a greedy algorithm obtains a solution up to a provable approximation factor [6]. The algorithm initializes $\tilde{N}_{\text{diverse}}$ to the empty set, and at each step adds a new set $y \in \tilde{N}_{\text{candidates}}$ to it which minimizes $\max_{z \in \tilde{N}_{\text{diverse}}} |y \cap z|$. The constant $c$ is set to 3 in order to show the users a sufficiently diverse and yet not overwhelming number of candidates in the SAG.

## 4.2   Patch Deletion to Build the SAG

After we have obtained the diverse set of candidates $\tilde{N}_{\text{diverse}}$, it is straightforward to build the SAG. Each element of $\tilde{N}_{\text{diverse}}$ forms a root node for the SAG. Child nodes are recursively generated by deleting one patch at a time from a parent node (equivalent to obtaining leave-one-out subsets of a parent set). We calculate the confidence of each node by a forward pass of the image represented by the node through the deep network. Since nodes with low probability represent less useful sets of patches, we do not expand nodes with probability less than a threshold $P_l$ as a measure to avoid visual clutter in the SAG. $P_l$ is set to $40\%$ as a sufficiently low value.

A flowchart illustrating the steps involved to generate a SAG for a given image input is shown in Fig. 4. All the SAGs presented in the paper explain the predictions of VGGNet [27] as the classifier. Results on ResNet-50, as well as details regarding the computation costs for generating SAGs are provided in the appendix.

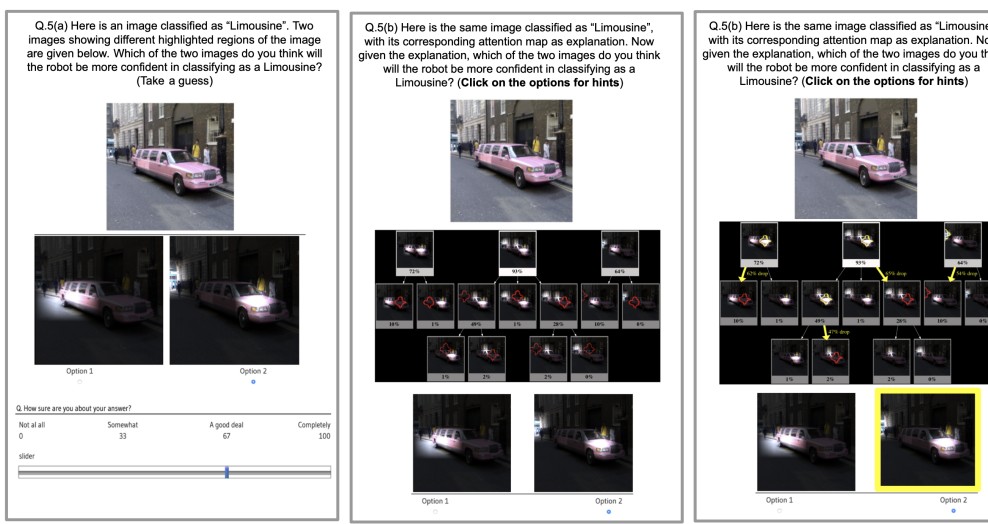

(a) Before explanation shown      (b) After explanation shown      (c) Clicking on one of the options

Figure 5: An example question used in the user study: (a) first, two options presented without a SAG explanation; (b) then, the same two options presented but now with a SAG explanation; (c) same as (b), but when a participant clicks on one of the options, related parts in the SAG are highlighted.

## 5 User Study

We conducted a user study to evaluate the effectiveness of our proposed SAG visualization.[2] User studies have been a popular method to evaluate explanations. For instance, Grad-CAM [24] conducted a user study to evaluate faithfulness and user trust on their saliency maps, and LIME [21] asked participants to predict generalizability of their method by showing their explanations to the participants. This section describes the design of our study and its results.

### 5.1 Study Design and Procedure

We measured human *understanding* of classifiers indirectly with *predictive power*, defined as the capability of predicting $f_c(N)$ given a new set of patches $N \subset x$ that has not been shown. This can be thought of as answering *counterfactual* questions – "how will the classification score change if parts of the image are occluded?" Since humans do not excel in predicting numerical values, we focus on answering *comparative queries*, which predict the TRUE/FALSE value of the query: $\mathbb{I}(f_c(N_1) > f_c(N_2))$, with $\mathbb{I}$ being the indicator function. In other words, participants were provided with two new sets of patches that have not been shown in the SAG presented to them and were asked to predict which of the two options would receive a higher confidence score for the class predicted by the classifier on the original image. Using this measure, we compared SAG with two state-of-the-art saliency map approaches I-GOS [20] and Grad-CAM [24].

We recruited 60 participants comprising of graduate and undergraduate students in engineering students at our university (37 males, 23 females, age: 18-30 years). Participants were randomly divided into three groups with each using one of the three saliency map approaches (i.e., between-subjects study design). They were first shown a tutorial informing them about the basics of image classification and saliency map explanations. Then they were directed to the task that involved answering 10 sets of questions. Each set involved an image from ImageNet. These 10 images are sampled from a subset of ImageNet comprising of 10 classes. Each question set composed of two sections. First, participants were shown a reference image with its classification but no explanation. Then they were asked to select one of the two different perturbed versions of the reference image with different regions of the image occluded , based on which they think would be more likely to be classified as the same class as the original image (shown in Fig. 5(a)). They were also asked to provide a confidence rating about how sure they were about their response. In the second section, the

---

[2]The user study has been approved by IRB, followed the informed consent procedure and involved no more than minimal risk to the participants.

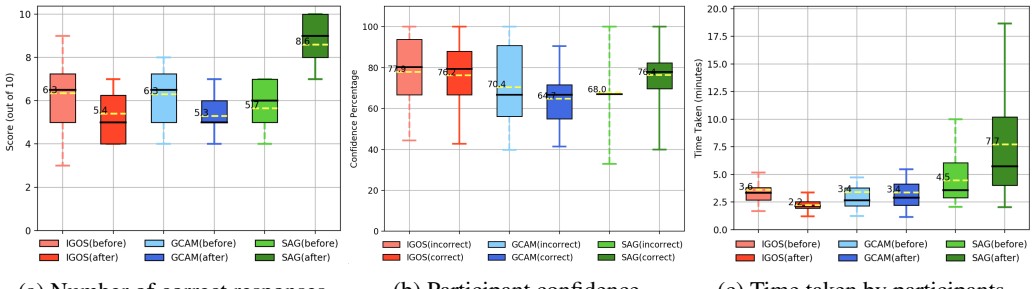

| (a) Number of correct responses | (b) Participant confidence | (c) Time taken by participants |

Figure 6: User study results comparing SAG to I-GOS and Grad-CAM. Dashed yellow lines in the box plots denote mean values.

participants were shown the same reference image, but now with a saliency map or SAG additionally. They were asked the same question to choose one of the two options, but this time under the premise of an explanation. Along with a SAG representation, they can click on an option to highlight the corresponding SAG nodes that have overlapping patches with the selected option and also highlight their outgoing edges (as shown in Fig. 5(c)). Each participant was paid $10 for their participation.

The metrics obtained from the user study include the number of correct responses among the 10 questions (i.e., score) for each participant, the confidence score for each of their response (i.e., 100 being completely confident; 0 being not at all), and the time taken to answer each response.

## 5.2 Results

Fig. 6 shows the results comparing the metrics across the three conditions. Fig. 6(a) indicates that participants got more answers correct when they were provided with SAG explanations (Mean=8.6, SD=1.698) than when they were provided with I-GOS (Mean=5.4, SD=1.188) or Grad-CAM (Mean=5.3, SD=1.031) explanations. The differences between SAG and each of the two other methods are statistically significant ($p <$0.0001 in Mann-Whitney U tests for both[3]).

Fig. 6(b) shows the participants' levels of confidence for correct and incorrect answers across all three conditions after being provided with the explanations. The plots show that their confidence levels are almost the same for both correct and incorrect responses in the cases of I-GOS and Grad-CAM. However, for the case of SAG, participants have lower confidence for incorrect responses and higher confidence for correct responses. Interestingly, the variance in confidence for incorrect answers is very low for the participants working with SAG explanations. The increased confidence for correct responses and reduced confidence for incorrect responses implies that SAG explanations allow users to "know what they know" and when to trust their mental models. The indifference in confidence for correctness in I-GOS and Grad-CAM may imply that participants lack a realistic assessment of the correctness of their mental models.

Fig. 6(c) shows that SAG explanations required more effort for participants to interpret explanations. This is expected because SAGs convey more information compared to other saliency maps. However, we believe that the benefits of gaining the right mental models and "appropriate trust" justify the longer time users need to digest the explanations.

## 5.3 Ablation Study

The two major components of the SAG condition used in the study are the graph-based attention map visualization and the user interaction for highlighting relevant parts in the visualization. As an ablation study, we include two ablated versions of SAGs: (1) SAG/I, which is a SAG without the click interaction, comprising only of the graph visualization and (2) SAG/G, which is a SAG without the graph visualization, comprising only of the root nodes and the interaction. These root nodes of the SAG are similar in spirit to the if-then rules of Anchors [22] and serve as an additional baseline.

---

[3]To check the normality of each variable, we first ran Shapiro-Wilk tests, and the results indicated that some of the variables are not normally distributed. Thus we used the Mann-Whitney U tests which are often used for comparing the means of two variables that are not necessarily normally distributed [30].

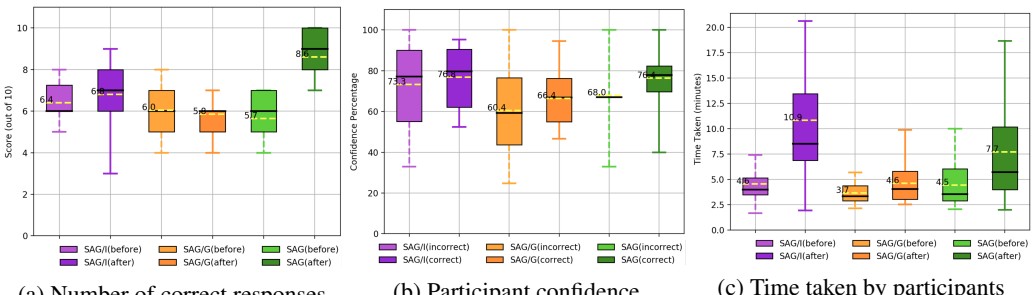

| (a) Number of correct responses | (b) Participant confidence | (c) Time taken by participants |

Figure 7: Ablation study results comparing SAG to SAG/I and SAG/G.

To evaluate how participants would work with SAG/I and SAG/G, we additionally recruited 40 new participants (30 males, 10 females, age: 18-30 years) from the same recruitment effort as for earlier experiments and split them into two groups, with each group evaluating an ablated version of SAGs via the aforementioned study procedure. The results of the ablation study are shown in Fig. 7. The participants received significantly lower scores when the user interaction (SAG/I) or the graph structure (SAG/G) are removed ($p < 0.0001$ in Mann-Whitney U tests for both; data distribution shown in Fig. 7a). This implies that both the interaction for highlighting and the graph structure are critical components of SAGs. The correlations of high confidence with correctness and low confidence with incorrectness are maintained across the ablated versions (as in Fig. 7b). Participants spent a longer time to interpret a SAG when they were not provided with the interaction feature, while interpreting just the root nodes took a shorter time (as in Fig. 7c). It is also worth noting that the differences between SAG without the interactive feature (SAG/I) and each of the two baseline methods (i.e., Grad-CAM and I-GOS) are also statistically significant ($p = 0.0004$ and $p = 0.0012$, respectively), showing the effectiveness of presenting multiple explanations using the graph structure. More data for all the 100 participants involved in the studies is provided in the appendix.

## 6   Conclusions and Future Work

In this paper, we set out to examine the number of possible explanations for the decision-making of an image classifier. Through search methods, especially beam search, we have located an average of 2 explanations per image assuming no overlap and 5 explanations per image assuming an overlap of at most 1 patch (about 2% of the area of the image). Moreover, we have found that 80% of the images in ImageNet has an explanation of at most 20% of the area of the image, and it is shown that beam search is more efficient than other saliency map approaches such as GradCAM and I-GOS in locating compact explanations at a low resolution.

Based on these findings, we presented a new visual representation, SAG, that explicitly shows multiple explanations of an image. It effectively shows how different parts of an image contribute to the confidence of an image classifier's decision. We conducted a large-scale human-subject study (i.e., 100 participants), and participants were able to answer counterfactual-style questions significantly more accurately with SAGs than with the baseline methods.

There are many interesting future research directions. One weakness of our approach is that it takes more time for people to digest SAGs than the existing methods. This could be mitigated via more advanced interfaces that allow users to interactively steer and probe the system to gain useful insights [12]. Another direction is to generalize our approach to multiple images and apply our methodology to other modalities such as language and videos.

## Acknowledgements

This work was supported by DARPA #N66001-17-2-4030 and NSF #1941892. Any opinions, findings, conclusions, or recommendations expressed are the authors' and do not reflect the views of the sponsors.

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
