# 7 Appendix

## 7.1 User Study Data

Here we provide the scores of all the 100 users that participated in our user study. We see that the scores are fairly random when participants are not provided with any explanation. Moreover, participants spending more time on the questions do not necessarily achieve higher scores.

After providing the explanations, we see that high scores (8 and above) are exclusively obtained by participants working with SAG and its ablations. As discussed earlier, participants working with SAG and SAG/I tend to have a higher response time than participants working with other explanations.

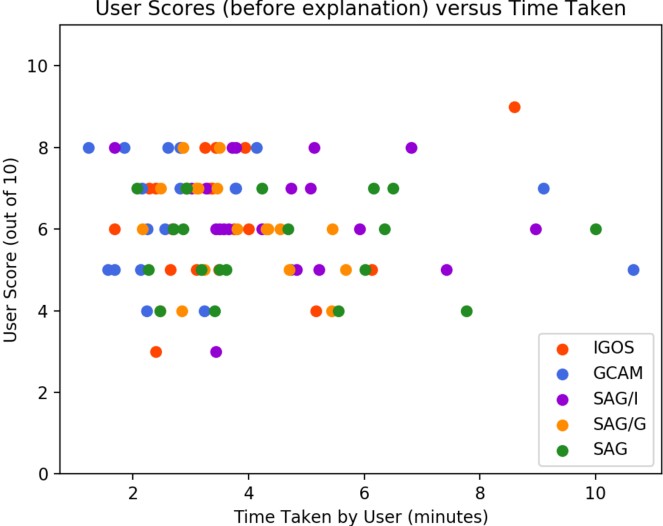

Figure 8: Performance versus Time scatter plot of all users before they are shown the explanations.

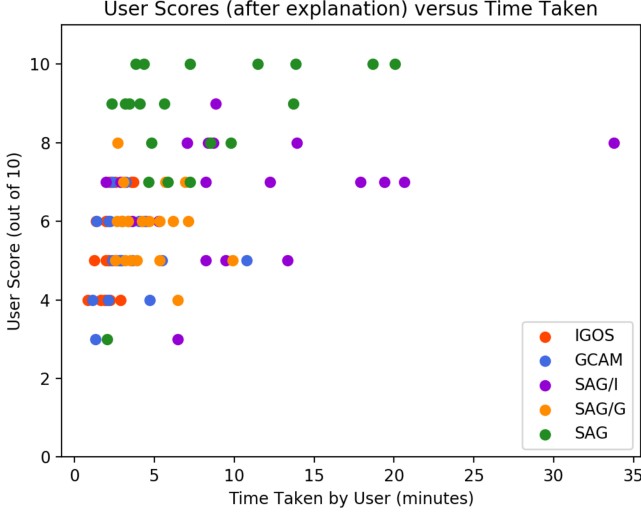

Figure 9: Performance versus Time scatter plot of all users after they are shown the explanations.

### 7.2 Minimum Sufficient Explanations: Analysis for ResNet

All the experiments and results in the paper use VGGNet as the black-box classifier. In this section of the appendix, we provide a brief analysis of the nature of multiple minimal sufficient explanations (MSEs) for ResNet [11] as the black-box classifier. We use the same dataset i.e., the ImageNet validation set for these experiments.

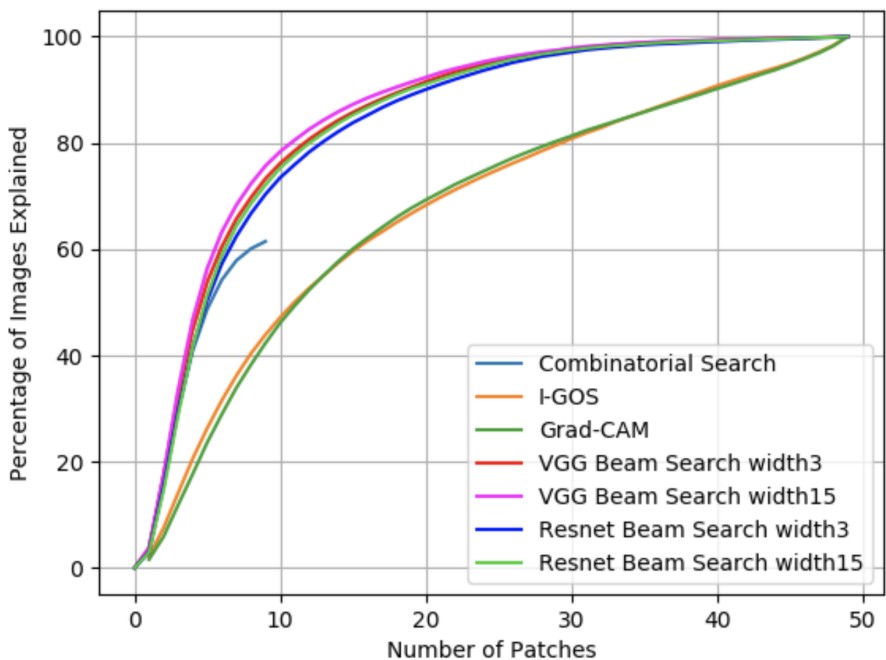

Figure 10: Percentage of images explained by different number of patches: comparing ResNet and VGGNet.

| Method | Overlap = 0 | | | Overlap = 1 | | |
|---|---|---|---|---|---|---|
| | Mean | Variance | Mode | Mean | Variance | Mode |
| BeamS-3 (VGGNet) | 1.72 | 0.98 | 1 | 4.18 | 7.24 | 2 |
| BeamS-3 (ResNet) | 1.66 | 2.61 | 1 | 3.27 | 5.49 | 1 |
| BeamS-15 (VGGNet) | 1.87 | 1.01 | 2 | 4.51 | 7.22 | 3 |
| BeamS-15 (ResNet) | 1.73 | 2.62 | 1 | 3.41 | 5.94 | 1 |

Table 3: Number of diverse MSEs obtained by allowing for different degrees of overlap: comparing ResNet and VGGNet

From Fig. 10, we see that the beam search is slightly sub-optimal at finding minimal MSEs for ResNet than for VGGNet. Similarly, Table 3 shows that beam search finds a lower number of MSEs on average when the classifier being explained is ResNet. The difference between the modes of the distributions for the two classifiers becomes stark on increasing the beam width. We hypothesize that these differences in the two distributions for the number of MSEs are due to the different perturbation maps obtained for the two classifiers, which we use for guiding the beam search. Digging deeper into the nature of MSEs for various classifiers is one of the possible avenues for future research.

### 7.3 Computation costs

A representation of computation cost of all the methods and baselines used in our work is provided in table 4 in terms of the wallclock time taken by each method to find and build the explanation for a single image. These values were obtained over a random sample of 100 images from the IMAGENET validation set using a single NVIDIA Tesla V100 GPU.

| Method | Time taken to find the explanation (T1) | | Time taken to build the explanation (T1 + time taken to create the visualization) | |
|---|---|---|---|---|
| | Mean | Variance | Mean | Variance |
| GradCAM | 3.10 | 0.32 | 3.66 | 0.38 |
| IGOS | 4.96 | 0.05 | 5.44 | 0.06 |
| CombS | 9.73 | 4.07 | 12.40 | 10.79 |
| BeamS-3 | 11.34 | 49.16 | 19.97 | 477.41 |
| BeamS-5 | 14.64 | 126.78 | 21.44 | 225.62 |
| BeamS-10 | 20.65 | 299.95 | 28.39 | 555.72 |
| BeamS-15 | 27.81 | 625.85 | 38.12 | 1855.36 |

Table 4: Computation times (in seconds) to find and build the explanation for a single image by various methods used in our work.

### 7.4 SAG Explanations for Wrong Predictions and Debugging

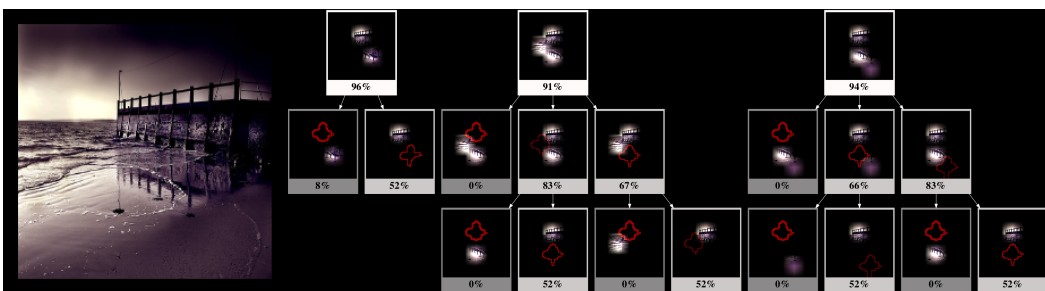

Figure 11: SAG explanation for the wrong classification of this image as "shopping cart". Correct class is "seashore".

SAGs can be particularly useful to gain insights about the predictions of a neural network and facilitate debugging in case of wrong predictions. For example, Fig. 11 shows that the image with ground truth class as "seashore" is (wrongly) classified as a "shopping cart" by VGG-Net because the coast fence looks like a shopping cart. Interestingly, the classifier uses the reflection of the fence as further evidence for the class "shopping cart": with both the fence and the reflection the confidence is more than 83% but with only the fence it was 52%. The patch corresponding to the reflection is not deemed enough on its own for a classification of shopping cart(evident from the drop in probabilities shown in SAG).

We provide more examples of SAGs for explaining wrong predictions by VGG-Net. These SAG explanations provide interesting insights into the wrong decisions of the classifier. For contrast, we also show the corresponding Grad-CAM and I-GOS explanations for the wrong predictions.

### 7.4.1 Predicted class: *Pickup*; True class: *Limousine*

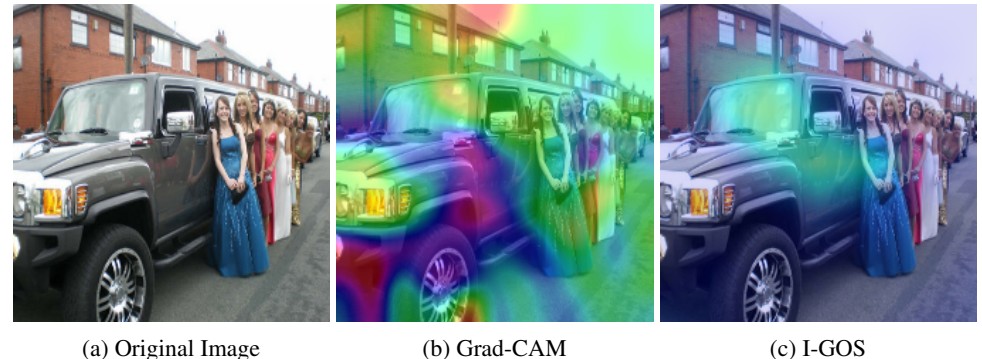

(a) Original Image        (b) Grad-CAM        (c) I-GOS

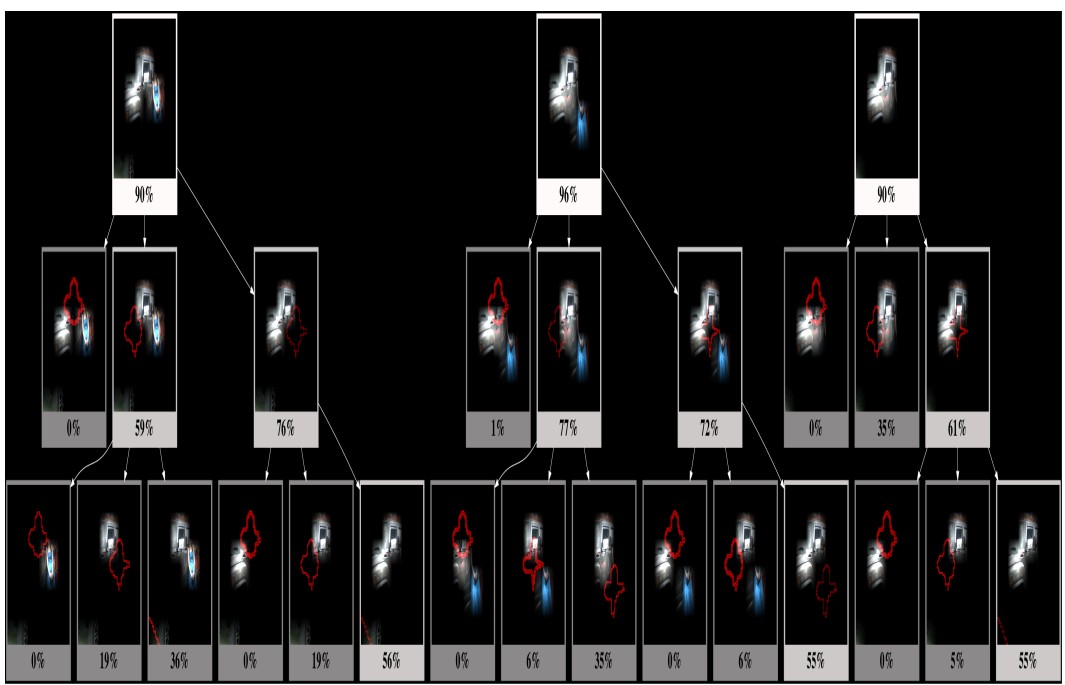

(d) SAG

### 7.4.2 Predicted class: *Golf-cart*; True class: *RV*

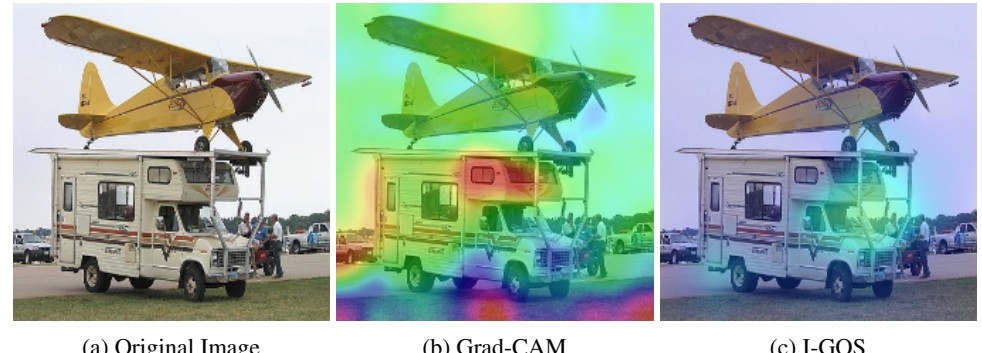

(a) Original Image         (b) Grad-CAM         (c) I-GOS

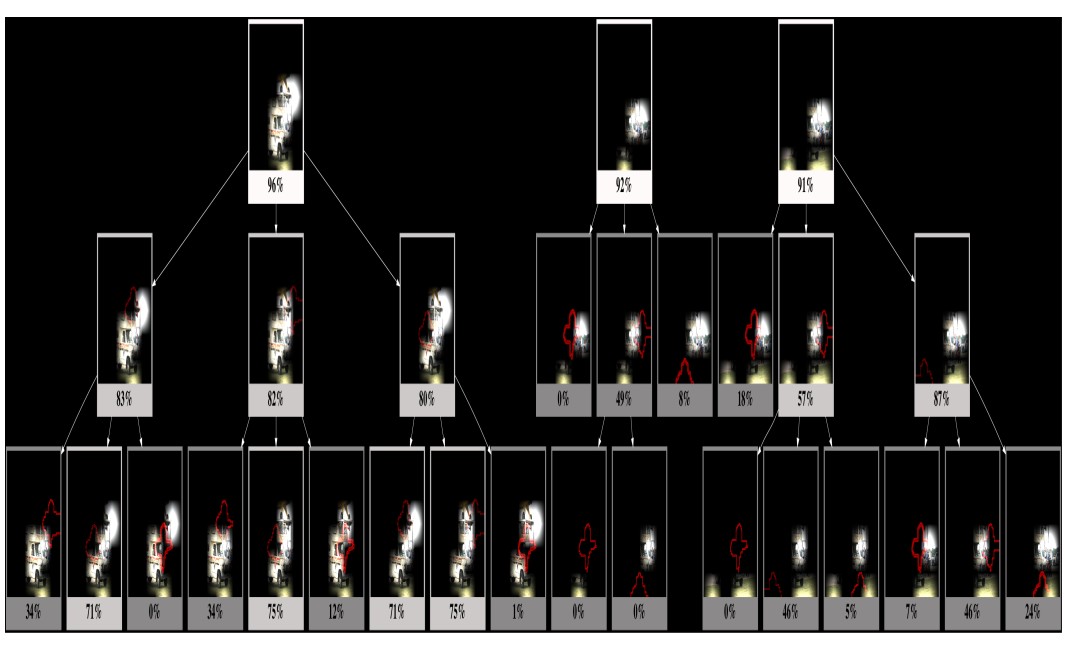

(d) SAG

### 7.4.3 Predicted class: *Prison*; True class: *Library*

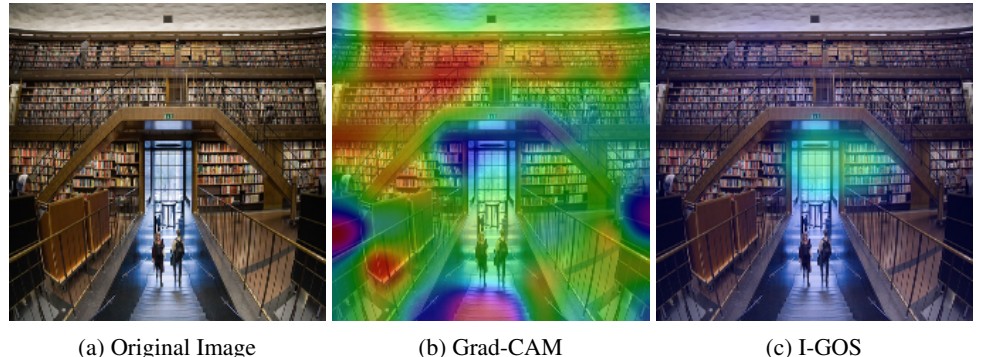

(a) Original Image          (b) Grad-CAM          (c) I-GOS

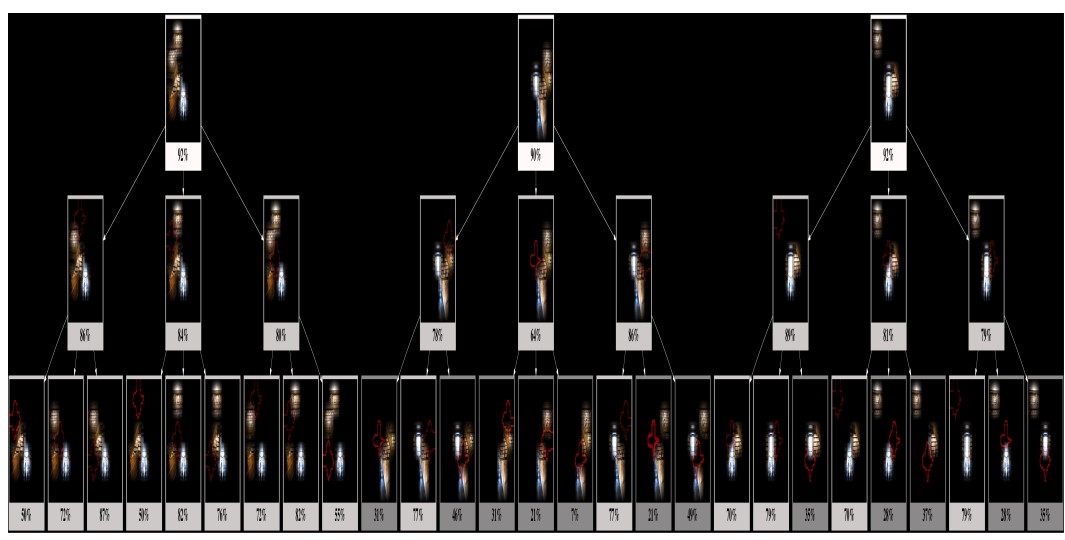

(d) SAG

### 7.4.4 Predicted class: *Shopping-cart*; True class: *Moped*

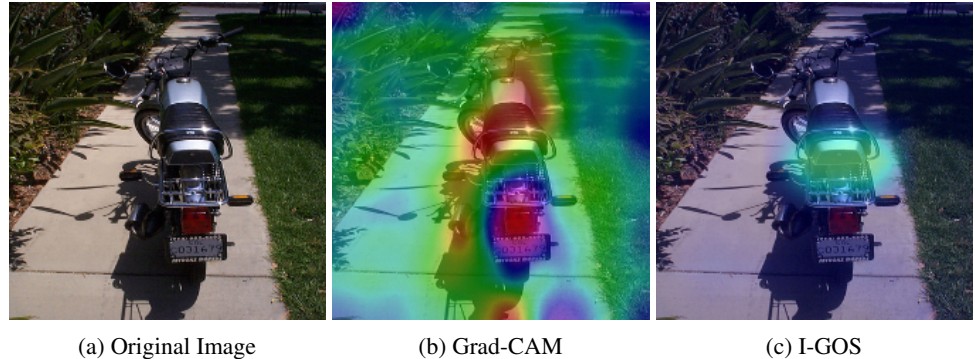

(a) Original Image      (b) Grad-CAM      (c) I-GOS

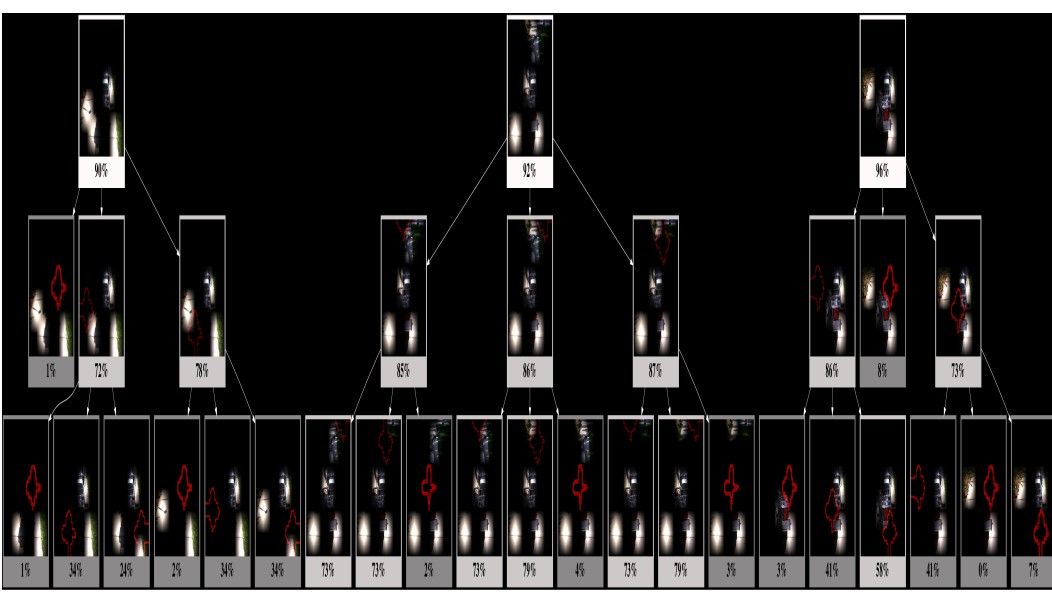

(d) SAG

### 7.4.5 Predicted class: *Space-shuttle*; True class: *Racecar*

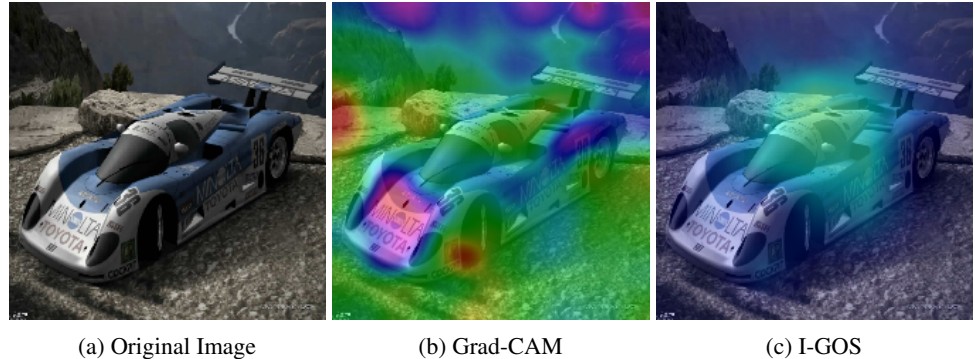

(a) Original Image    (b) Grad-CAM    (c) I-GOS

(d) SAG

### 7.5 SAG Examples

Here we provide more examples of SAGs for various images with their predicted (true) classes. In order to emphasize the advantage of our approach over traditional attention maps, we also provide the corresponding Grad-CAM and I-GOS saliency maps.

### 7.5.1 Class: *Goldjay*

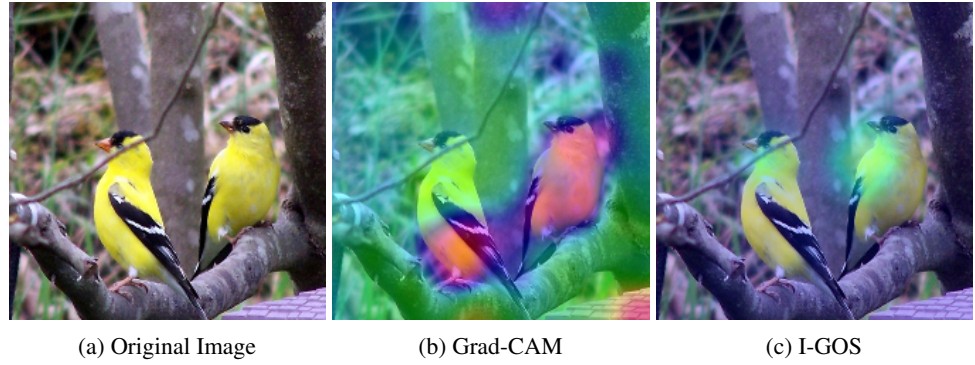

(a) Original Image      (b) Grad-CAM      (c) I-GOS

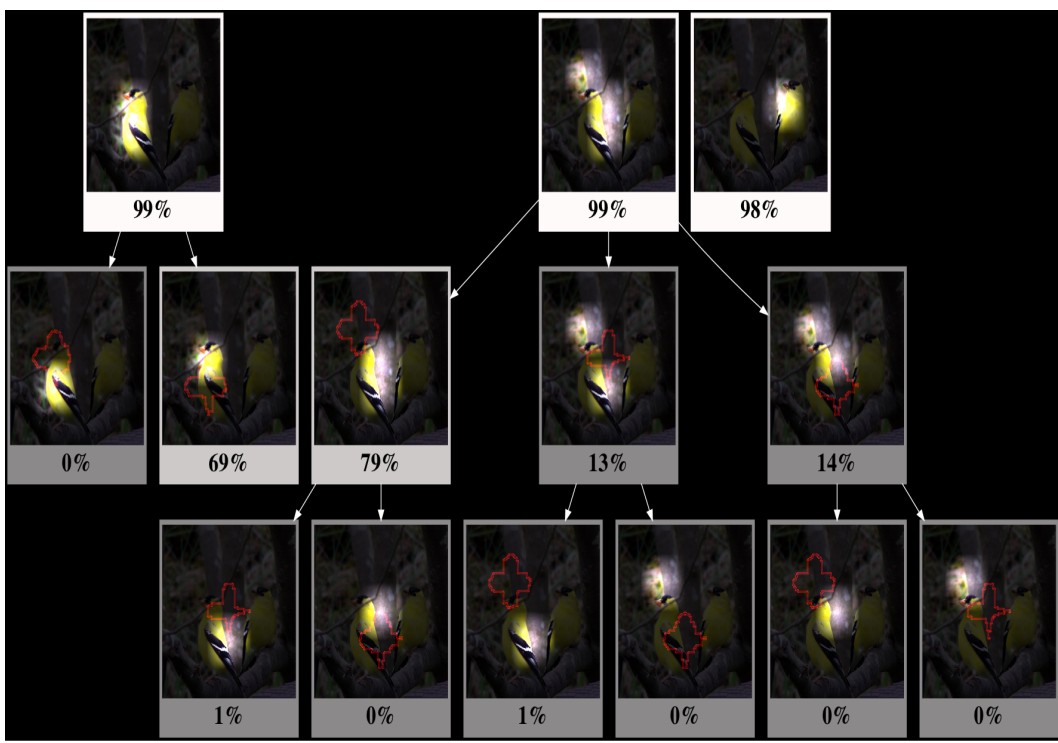

(d) SAG

**7.5.2   Class: *Schoolbus***

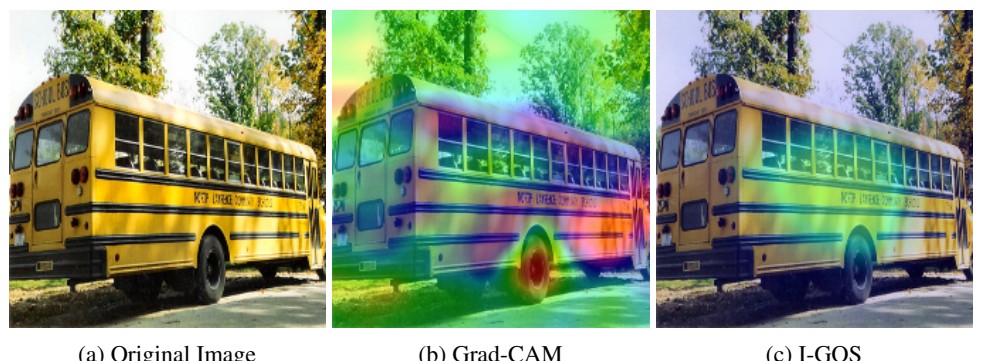

(a) Original Image          (b) Grad-CAM          (c) I-GOS

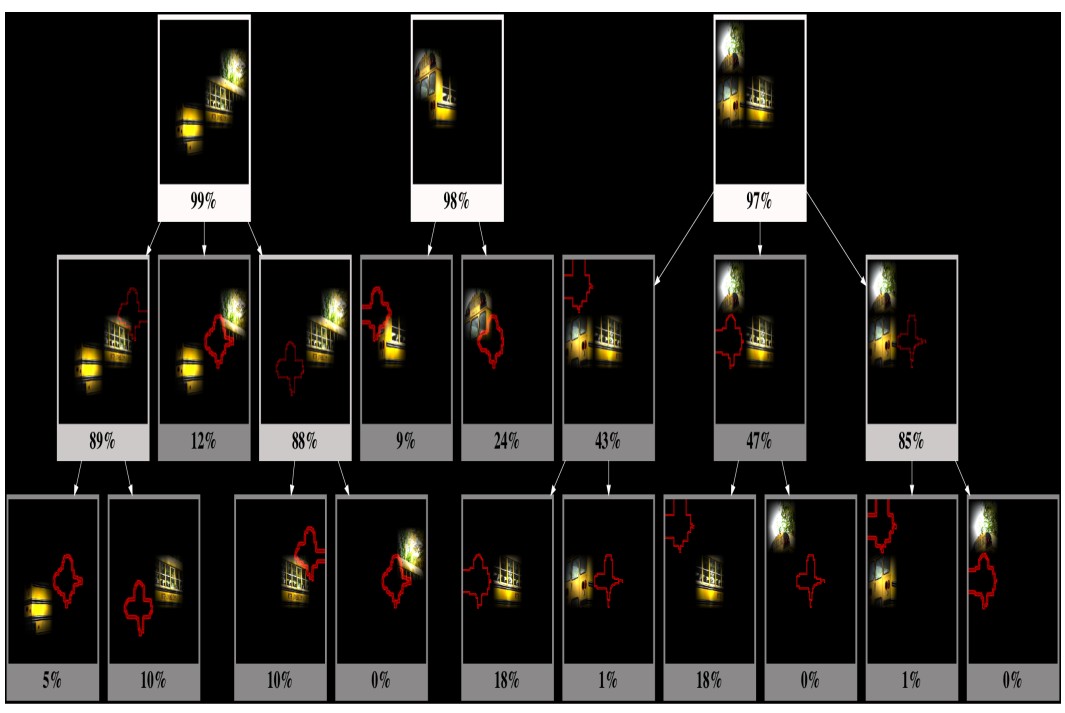

(d) SAG

### 7.5.3 Class: *Airedale Terrier*

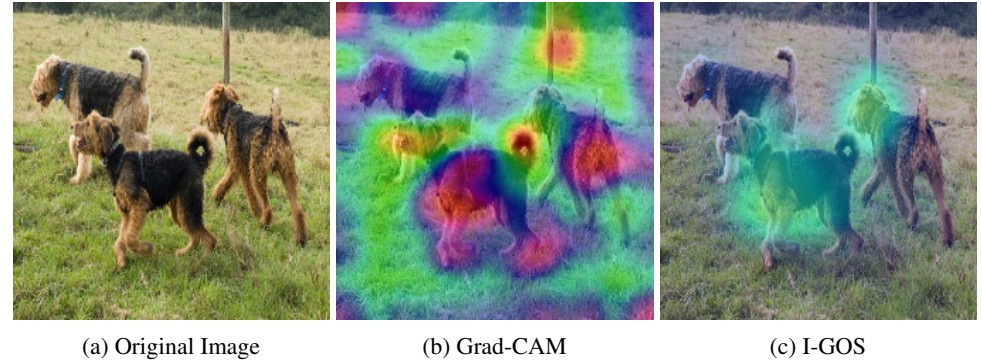

(a) Original Image      (b) Grad-CAM      (c) I-GOS

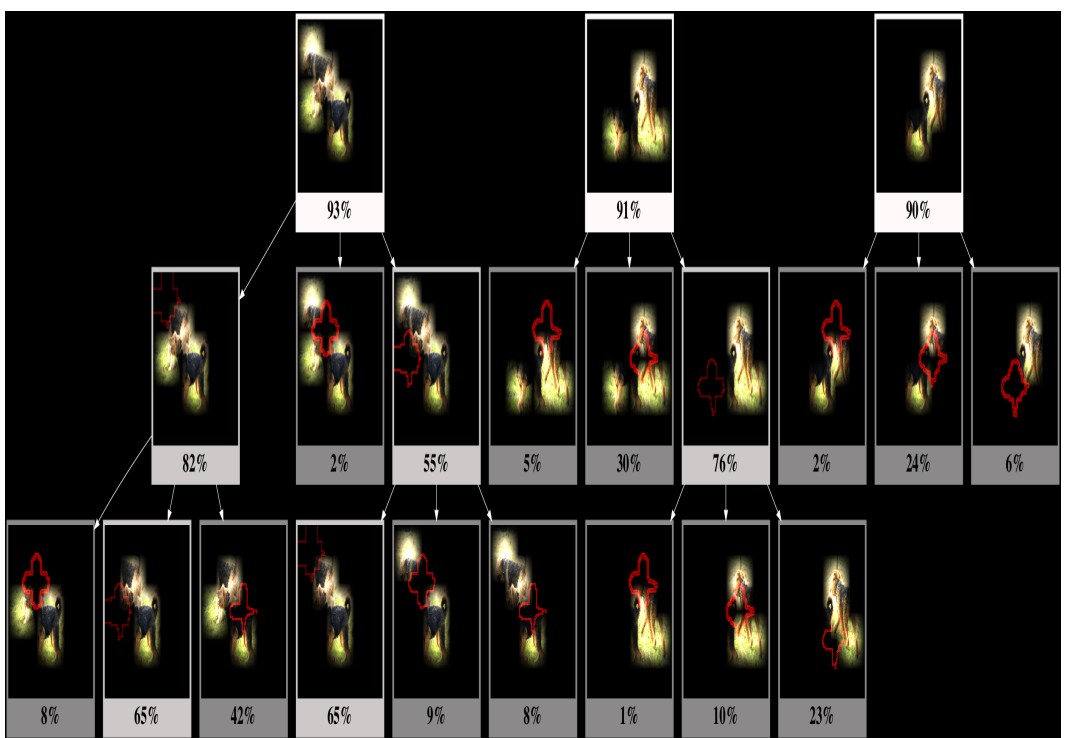

(d) SAG

### 7.5.4 Class: *Peacock*

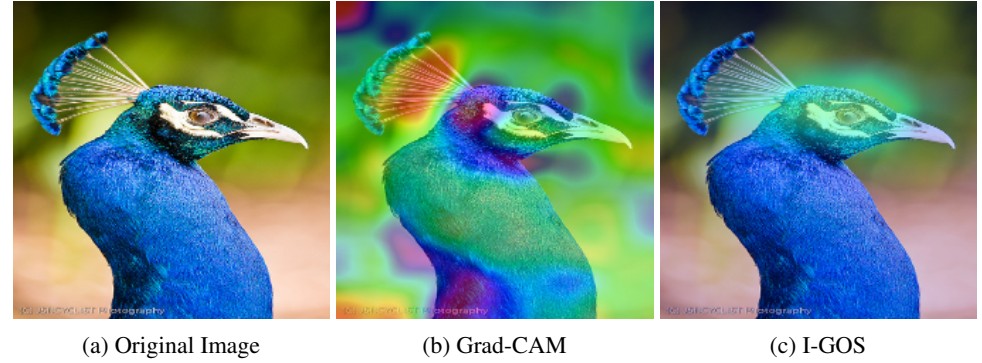

(a) Original Image        (b) Grad-CAM        (c) I-GOS

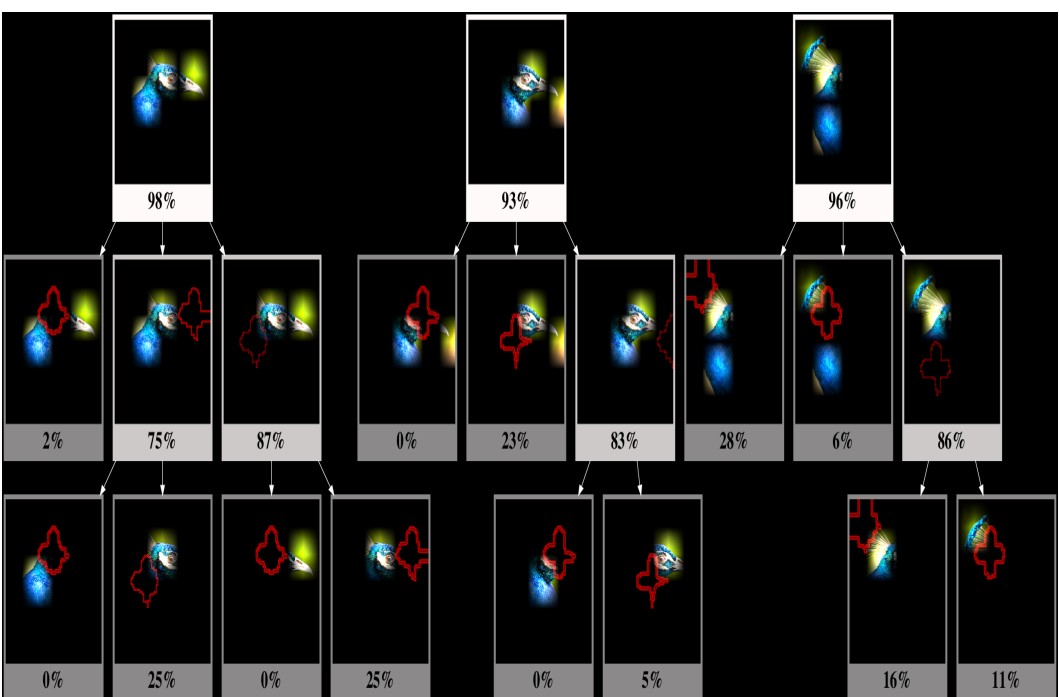

(d) SAG

**7.5.5 Class: *RV***

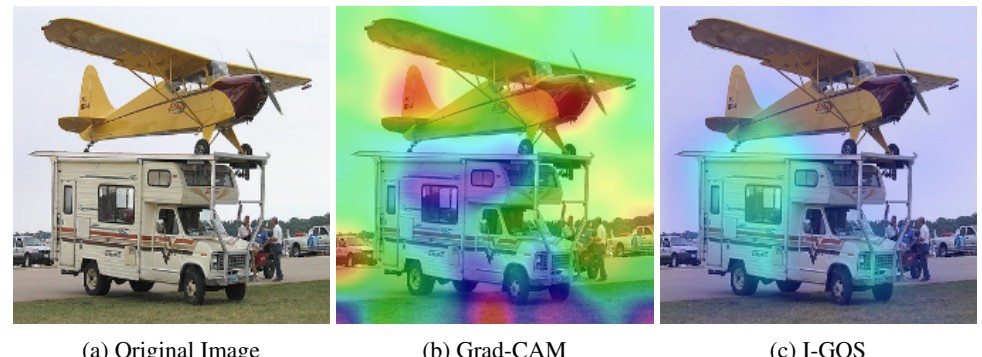

(a) Original Image      (b) Grad-CAM      (c) I-GOS

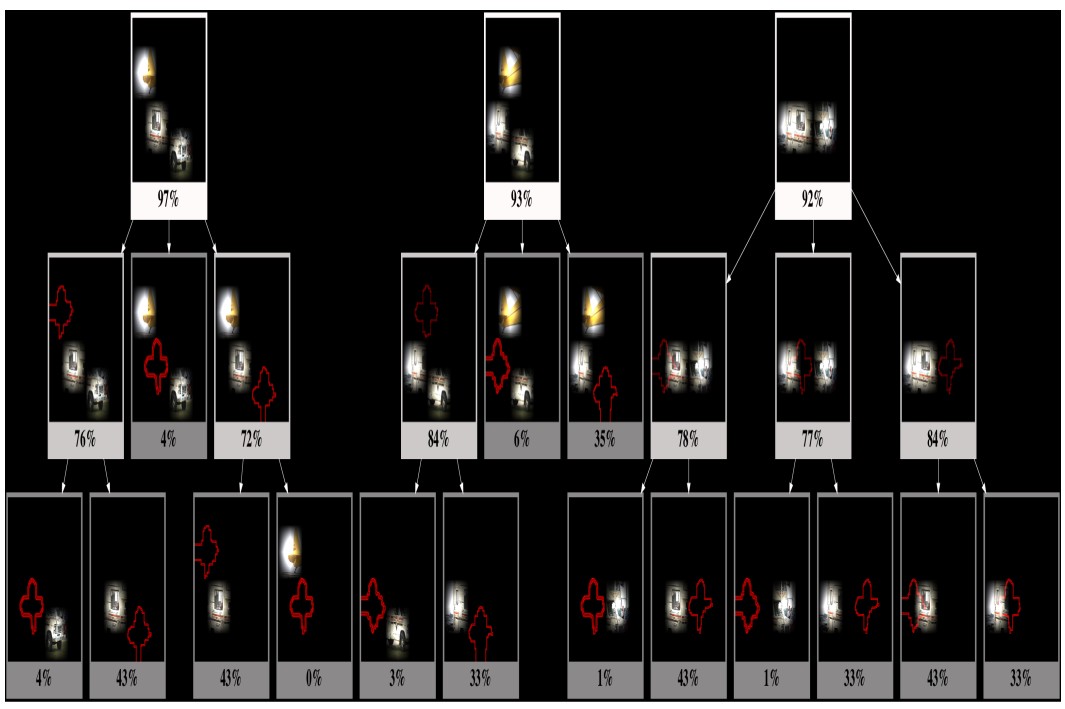

(d) SAG

### 7.5.6 Class: *Racecar*

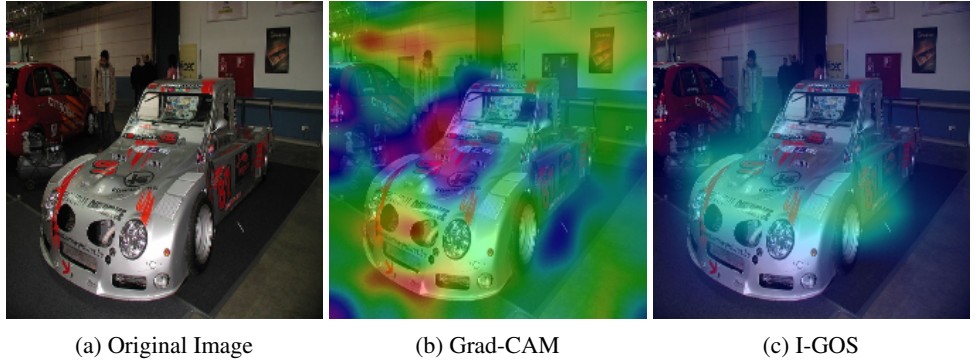

(a) Original Image       (b) Grad-CAM       (c) I-GOS

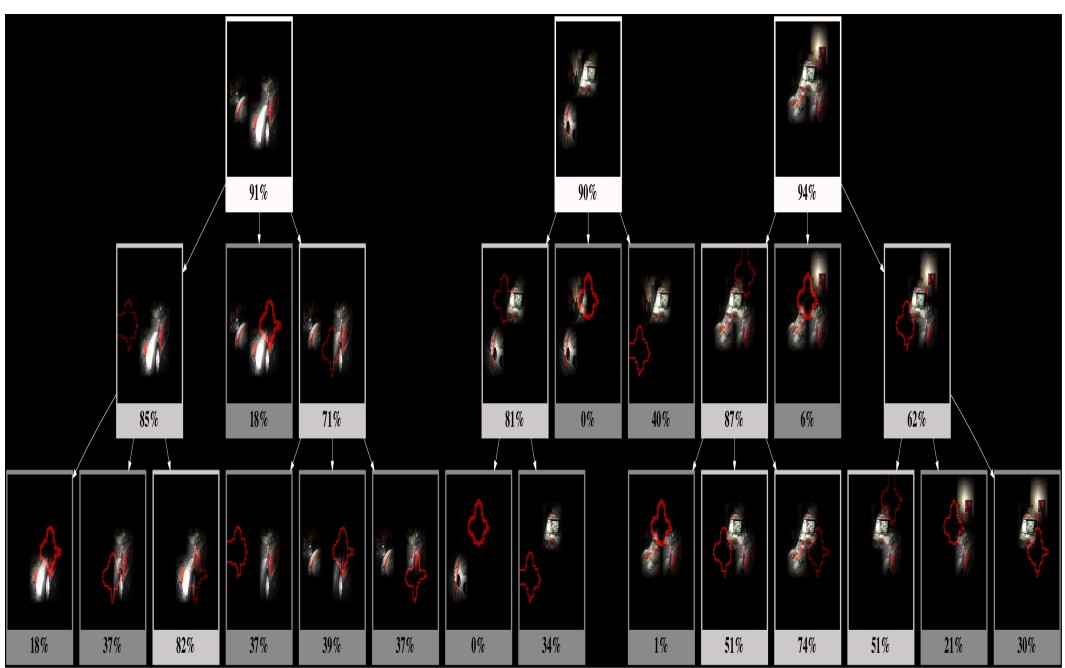

(d) SAG

### 7.5.7 Class: *Barbell*

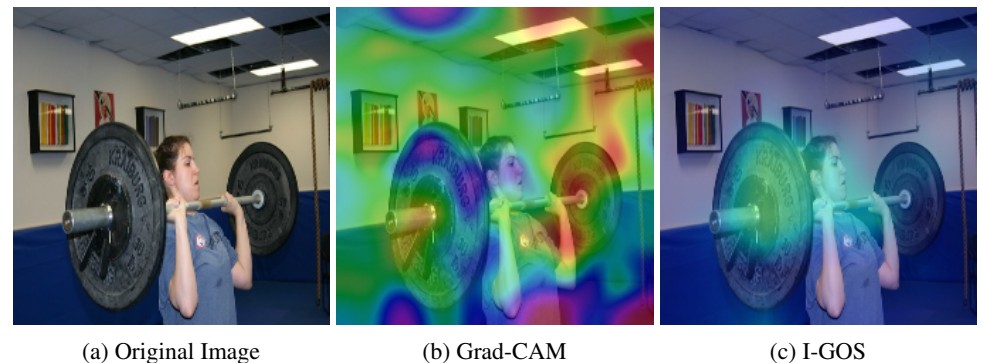

(a) Original Image          (b) Grad-CAM          (c) I-GOS

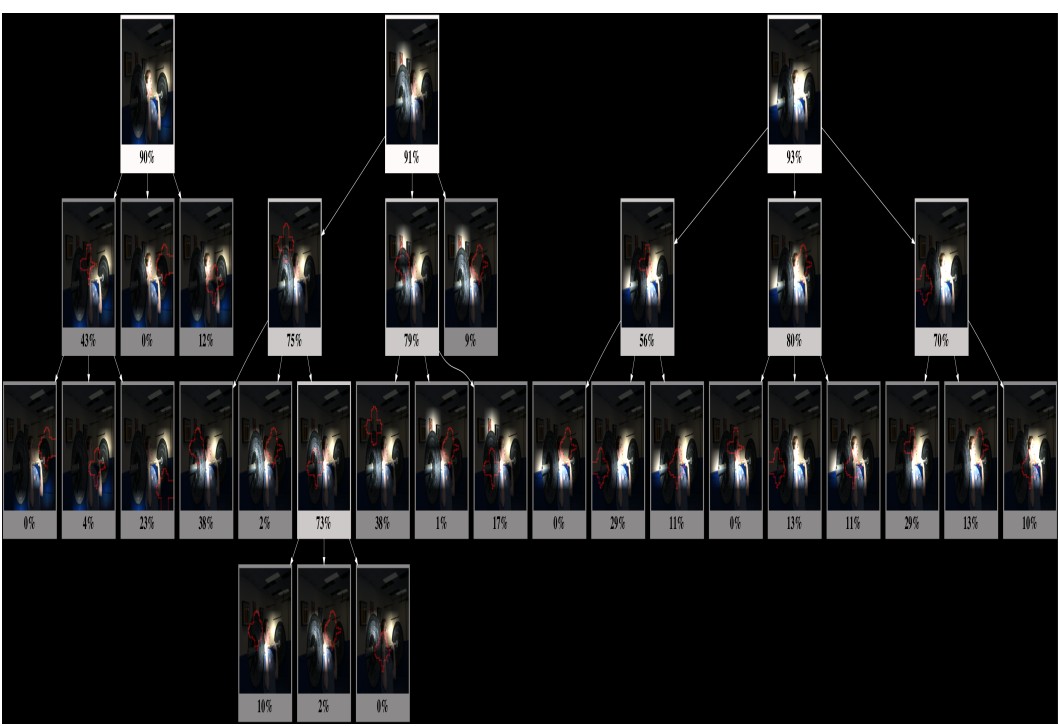

(d) SAG