# OpenReview forum: "One Explanation is Not Enough: Structured Attention Graphs for Image Classification"
_NeurIPS.cc/2021/Conference — NeurIPS 2021 Poster_

### Official Review · Reviewer_gy9K · 2021-06-28

**Rating:** 4
**Confidence:** 3

**Summary:**

The manuscript introduces Structured Attention Graphs (SAG) that presents a set of attention maps to show the sufficiency of different spatial regions in the input image for a deep classifier to output confident predictions. To evaluate whether users can utilize SAG to understand some certain behavior of the deep net, the authors conduct a user study experiment of SAG against two existing baseline explanations and show that SAG is a better explanation method.

**Limitations And Societal Impact:**

The authors have adequately addressed the limitations and potential negative societal impact of their work.

**Main Review:**

### Originality
The paper provides a new technique to visualize CNNs different from existing approaches. However, I have several concerns on the motivations of the paper.

1) The paper first argues that existing approaches are based on an assumption that their feature importance maps (a.k.a. attention maps in this paper) are sufficient to construct a *mental model*; however, as the definition of “mental model” is not formally introduced before its first use at line 29, it is not clear to me whether the authors’ criticism towards the prior work stands. Secondly, the first section introduces three questions (line 33-36) as the motivations for the rest of the paper but does not give more intuitions why these questions are necessary to answer and what can the community be benefited from by answering these questions. Furthermore, the authors mention “CNN decisions may be more fully explained…” at line 41 but do not intend to define what is “fully explained”.

2) I find the motivation of searching subregions (“... if the CNN is capable of achieving the same confidence from a subimage, then the rest of the image may not add substantially to the classification.” at line 113-114) is counterintuitive because it does not consider interactions between features. Consider a binary classifier $F(x_1, x_2) = sign[x_1 + x_1x_2]$. When $x_2=0$, $x_1 > 0$ is sufficient to output $F=1$ but when $x_2 < -1$, $x_1$ can negatively contribute to class 1. Therefore only visiting the case $x_2=0$ may not be enough to conclude whether $x_2$ is substantial to the classification.

### Quality
I have one question regarding the definition of MSE: why do we have thresholding in the definition of MSE $f_c(N_i) > P_hf_c(x))$ instead of maximizing the output confidence $f_c(N_i)$ ? The thresholding is the cause of having multiple subregions that exceed the threshold but why not find the subregion that maximizes the output -- namely the most sufficient subregion?

### Clarity
The paper uses a lot of hyper-parameters but does not intend to discuss how these parameters are chosen and how different choices of these hyper-parameters will influence the result. The lack of a sensitivity analysis of these parameters or justifications of the current choices (if an experiment is not necessary) is not helpful to support the conclusion the paper draws, especially when there are a lot of hyper-parameters used in this paper, including:
- $P_h=0.9$ at line 133
- $r=7$ at line 147
- $m=10$ at line 160
- $q=15$ at line 174
- “If they have less than two patches in common” at line 197
- “The constant c is set to 3 …” at line 242
- “$P_l$ is set to 40%” at line 250

I would suggest to provide more explanations to these hyper-parameters to improve the clarity of the work. Some choices may be obvious to the authors but less intuitive to readers.

### Significance
 The results are important and others (researchers or practitioners) are likely to use the ideas or build on them if the motivation of the work can be well-explained.


**Time Spent Reviewing:**

8

---

> ### Author Response · Authors · 2021-08-11
> **One Explanation is Not Enough: A Response to gy9K**
>
> Thank you for your thoughtful comments.
>
> “the definition of ‘mental model’ is not formally introduced”
>
> Response: We are using the word “mental model” informally to refer to the human user’s internal model of the network that helps them predict the behavior of the network on different subimages. We believe that this reflects a higher level of understanding than only visualizing the activations on different parts of the network.  At the same time, it helps  evaluate the explanations quantitatively, putting explainable deep learning on more solid scientific ground. Our user study precisely evaluates how good this mental model is by testing the predictions of the users on the test data and compares them to the baseline methods. We will clarify it in our manuscript.
>
> “Does not give more intuitions why these questions are necessary …”
>
> Response: We will expand on the importance of each question in the final paper. The first question asks if there are small localized explanations for most images. If there aren’t, i.e., if most of the image is needed to correctly classify the image, the explanations are not informative and cannot help build trust. Note that most quantitative evaluation metrics of attention maps focus on the localization aspect (MoRF/LoRF (Samek et al. 2017), insertion/deletion (Petsiuk et al. 2018)) where they check whether removing/inserting a small saliency region pointed to by the attention maps would sway the CNN decision, but whether those small attention maps can be obtained, or for how many images small attention maps can be obtained have been an open question. The second question explores the presence of multiple explanations of an image which is the main contribution of the paper. With the exception of the Anchors approach (which we compared with), prior work so far has mostly ignored the reality of multiple explanations, thus failing to account for a major source of robustness of neural networks. The third question studies the engineering aspect of constructing and displaying multiple explanations, which is critical to make the claims operational and improve the users’  mental models of the network’s performance.
>
> (Samek et al. 2017) W. Samek et al. Evaluating the visualization of what a deep neural network has learned. IEEE TNNLS, 2017.
>
> (Petsiuk et al. 2018) V. Petsiuk et al. Rise: Randomized input sampling for explanation of black-box models. BMVC 2018.
>
> “authors mention ‘CNN decisions may be more fully explained’ … what is ‘fully explained'"?
>
> Response: Concretely, by “fully explained,” we mean that the explanations are sufficient to correctly predict the behavior of the network on any subimage of the original image. However, the emphasis here is on “more.” By “more fully” explaining, we mean that the multiple explanations help predict correctly over a larger subset of subimages than a single explanation.
>
> “motivation of searching subregions … is counterintuitive …”
>
> Response: In general,  the amount of search must increase in order to ensure that all interactions beyond some minimum size are discovered. The beam width in our search procedure provides such a knob. A more important question is what we see in practice. An important empirical observation we made is that the classifier’s confidence in the correct class usually grows monotonically with the exposed area of the image in most cases. There is interaction between the subimages, but as we expose more parts of the image, the score generally goes up, and almost never goes down.  The search algorithm exploits this “monotone’ property and stops when the score raises above a threshold. This also allows us to view the problem as learning monotone DNF expressions [lines 53--60].
>
> “why do we have thresholding …”
>
> Response: We don’t want the most sufficient subregion because typically it includes the entire object if not the entire image. It is not a useful explanation to say that something is a cat because of its whole body. We would like to infer if the network would have classified it as a cat if it had only seen its face, or if it had only seen its tail, and so on. What is surprising and interesting is that a) a small set of patches (MSE) is often sufficient for correct classification and b) there are often more than one MSE that are equally good.
>
> “suggest to provide more explanations to these hyper-parameters …”
>
> Response: Thank you for the suggestion. Some of the hyperparameters of the search algorithms (r, m) are set to some reasonable values and never changed because we could not afford a larger search space. r=7 was chosen because that’s the maximal size where we can still run a filtered combinatorial search in reasonable time. m=10 was due to the same reason. The results show that such a filtered combinatorial search is not as effective as beam search. Larger m would make the performance better but also exponentially slower. For q, we tried different values of q=3, 5, 10 and 15 and showed the results of all of them in Fig. 3. Some other parameters have their rationale explained in the paper. We will try to improve the discussion of the hyperparameters in the final version.

---

> > ### Comment · Reviewer_gy9K · 2021-08-16
> > **I appreciate the thoughtful response but some of my concerns are still not addressed.**
> >
> > I appreciate the authors' response that tries to address my questions. I believe that the authors will present a new version that justify their choices of hyper-parameters. The hyper-parameter part is not my biggest concern as most of them seem to be ad-hoc choices and should not affect the results a lot. I still have some questions about the motivation part and the message the authors aim to convey in this paper.
> >
> > **Motivation: Explanations should help to construct a reasonable mental model of the classification decision for the particular image. (line 29)**
> >
> > As the authors define the *mental model* as *human user’s internal model of the network that helps them predict the behavior of the network on different sub-images* in the response, **why human wants to predict the prediction of the network when the prediction result is already an input argument for most explanations, like GradCAM**? I can think of two cases that might be in the authors' mind:
> >
> > - Case One: if the authors want to show another use case that we only provide the user with the explanation without telling them towards what prediction this visualization explains and ask them to if this visualization makes sense, we seem to fall into a situation when we ask people whether an answer (the explanation) makes sense when we hide the question (the output class we are trying to explain) from them. That is, why an answer should still be treated as an answer when the question is unknown.
> >
> > - Case Two: on the other hand, if the authors wants to show whether explanations can help people to reason from the explanations to the model's prediction by this work, then to me this paper is measuring how close the explanations are to the human's prior (*mental model* in the authors' word). However, several related work [1, 2, 3] (these paper may not focus on GradCAM or I-GOS but the axioms and theories in these paper about a good explanation should not be limited to the particular method they develop) has emphasized that the goodness of explanations should focus on whether the explanations can faithfully capture the model's performance instead of the audience's prior. That is, if we have a faithful explanation and it looks bad such that we can not reason from the explanation to the model's prediction, this is actually not be interpreted as a bug of the explanation but more likely as that the model is not learning the correct concept.
> >
> > For either of the cases above (or a third case that better describes the motivation of this paper), if the authors can help us to understand for what reason we can say SAG is a better explanation and why does this particular reason define a good explanation I will increase my score.
> >
> > A minor point: if the authors consider a good explanation should be used for a human to construct a good proxy model that approximate the network's output in a local region then apparently GradCAM is not motivated to serve as a proxy model at the first place (I am not as familiar to I-GOS as I do for GradCAM so I will use GradCAM as an example and please correct me if I am wrong). Instead, as far as I know, the proxy use of explanations motivates several other explanation methods, e.g. the LIME [4] paper mentioned in this paper.
> >
> > [1] Yeh, C., Hsieh, C., Suggala, A.S., Inouye, D.I., & Ravikumar, P. (2019). On the (In)fidelity and Sensitivity of Explanations. NeurIPS.
> >
> > [2] Sundararajan, M., Taly, A., & Yan, Q. (2017). Axiomatic Attribution for Deep Networks. ICML.
> >
> > [3] K. Leino, S. Sen, A. Datta, M. Fredrikson and L. Li, "Influence-Directed Explanations for Deep Convolutional Networks," 2018 IEEE International Test Conference (ITC), 2018, pp. 1-8, doi: 10.1109/TEST.2018.8624792.
> >
> > [4] Ribeiro, M.T., Singh, S., & Guestrin, C. (2016). "Why Should I Trust You?": Explaining the Predictions of Any Classifier. Proceedings of the 22nd ACM SIGKDD International Conference on Knowledge Discovery and Data Mining.

---

> > > ### Author Response · Authors · 2021-08-16
> > > **We appreciate the opportunity to explain our motivation in more detail.**
> > >
> > > Thank you for your probing questions and the helpful references.  We appreciate the opportunity to explain our motivation in more detail.
> > >
> > > “Why humans want to predict the predictions of the network.”
> > >
> > > The role of the explanation in our view is to enable the humans to develop a good (highly accurate) mental model.  In general a “model” of something is expected to behave like the real thing. However, a model is not a replica and is not expected to be an exact simulation. Hence we want a good mental model to help us correctly answer relevant questions about the deep network. In this work, we postulate that the relevant questions are comparative scores of the classifier on different subimages of the image. These questions provide a reasonable way to evaluate the human’s qualitative understanding of the network’s behavior on different subimages.
> > >
> > > In principle, it is possible that a human, after being shown I-GOS or Grad-CAM images, acquires such a mental model. After all, these visualizations purport to show which parts of the image are important for the network’s decisions. However, what we show through the user studies is that this is empirically not the case for the kind of questions we are interested in. The visualizations of I-GOS or Grad-CAM are not strong (informative) enough to allow the users to answer counterfactual questions about how different subimages are going to be classified. What this shows is that the “explanations” offered by I-GOS and Grad-CAM are at best incomplete in the sense of not being able to explain the network’s performance at the level of detail needed to answer our comparative questions. A key reason for this is that neither of these visualizations acknowledge the existence of multiple explanations.
> > >
> > > “Case One: if the authors want to show another use case that we only provide the user with the explanation without telling them towards what prediction...”
> > >
> > > This is not our setting. In all cases, the user knows the classification made by the network on the full image as well as its explanation. The user’s prediction is about comparative scores between two different subimages of the original image, which are themselves not in the SAG. This prediction is important because it sheds light on the correctness of the user's mental model which in turn is a function of the explanation. If one explains that a network classifies a cat as a cat because of its head rather than its tail, that claim can be verified by comparing what the network predicts when it is exposed to only its head vs. only its tail.
> > >
> > > “Case Two: on the other hand, if the authors wants to show whether explanations can help people to reason from the explanations to the model's prediction by this work…..has emphasized that the goodness of explanations should focus on whether the explanations can faithfully capture the model's performance instead of the audience's prior.”
> > >
> > > We agree that the user’s prior model about how a given image ought to be classified is not relevant, because it may not be faithful to the network’s predictions. This is not our setting.  By user’s “mental model,” we mean their model of what the network does, not their prior model of how to recognize the underlying class.
> > >
> > > “...the proxy use of explanations motivates several other explanation methods, e.g. the LIME [4] paper mentioned in this paper”
> > >
> > > We also do not view our work as providing “proxy prediction models” as in LIME and other systems. The SAGs cannot be used to independently predict the class of a new test image. They simply visualize the predictions of the original network on different subimages of the test image as a structured graph.  The goal of explanation is to provide visualizations that deepen the human user’s understanding of the classifier’s behavior, which is evaluated by asking them comparative counterfactual questions. We believe that our overall framework is uniquely useful to evaluate AI explanations and can be generalized to other kinds of questions.

---

### Official Review · Reviewer_RJDW · 2021-07-13

**Rating:** 6
**Confidence:** 4

**Summary:**

This paper states that a single saliency map is not enough and proposes to search for multiple explanations for each sample via beam search. A SAG module is introduced to visualize the combination of different attention maps.

**Limitations And Societal Impact:**

See main review.

**Main Review:**

#Originality: The paper presents an interesting perspective about explanation maps. Most of the previous work was looking for a single (as small as possible) explanation, but this paper found that there may be multiple similar explanations in a single sample.

#Quality:

1. In user study, SAG is compared with other single attention map approaches. I'm concern about the fairness, as you show multiple root SAG nodes and sub-nodes to users (shown in Figure 4) each time (more information is shown), if you also conduct similar operations on Grad-CAM (remove part of the map and observe the score drop), will it be better? Besides, it seems to be unfair to evaluate two maps when they are of different size.

2. What is the efficiency (time-costing) compared with other methods? Meanwhile, it costs more time for users to make a decision, why this happens?

#Clarity: The paper is well-written and easy to follow.

#Significance: The motivation is great, and this paper tries to solve a problem that has not been well studied yet. The conclusion shows some insight for later works.

**Time Spent Reviewing:**

2

---

> ### Author Response · Authors · 2021-08-11
> **One Explanation is Not Enough: A Response to RJGW**
>
> Thank you for your thoughtful comments.
>
> “if you also consider similar operations on Grad-CAM … will it be better? … seems to be unfair”
>
> Response: The contribution of our paper is precisely to provide an approach for producing multiple explanations and to show the value of multiple explanations (root SAG nodes and sub-nodes). What we have been showing the users is the way Grad-CAM is normally visualized. Developing an alternative as you suggested is an interesting direction, but different from the original Grad-CAM visualization and would be a novel contribution on its own. Besides, such an alternative would not have branches but only be a single set of heatmaps, unable to capture as much information as SAG does. Note that we also compared SAG to a baseline version with only showing multiple root nodes (SAG/G in Fig. 6), which shows better results than GradCAM but still worse than SAG.
>
> “efficiency compared to other methods? … more time for users to make decisions …”
>
> Response: SAG takes up to 30 seconds (ln. 251-253) for each image but for most images the time spent is significantly less than 30 seconds. We will report the average processing times in the final version. Naturally, SAGs need more time because of the search involved in computing MSEs unlike Grad-CAM and IGOS.  It costs more time for users to process data mainly because there is more information for users to digest. Answering questions correctly requires pooling information from different parts of the SAGs and combining the results, which is non-trivial.

---

> > ### Comment · Reviewer_RJDW · 2021-08-20
> > **Thanks for response**
> >
> > I really appreciate the response from the author, which solves most of my concerns.  At the same time, I refer to the opinions of other reviewers. I prefer to increase my score from 5 to 6.

---

### Official Review · Reviewer_cwRt · 2021-07-14

**Rating:** 7
**Confidence:** 4

**Summary:**

In this work the authors propose an interesting approach (SAG) in which the explanation of CNN classification networks in from of an attribution map is restructured and to the user visualised in a structured graph. In the graph, the influence of the region on the confidence of the classifier is shown in addition to the explanation. The authors demonstrate that their proposed approach is increasing the explanatory power compared to single heatmaps.

**Limitations And Societal Impact:**

The authors provide sufficient information on the limitations of their work.

**Main Review:**

The paper is generally well written and appears sound.

Instead of providing another novel XAI method, the authors address the important question of how to present explanations to the user. To this end, the authors first investigate if multiple/different explanations of one image are actually necessary and present a search method to find the minimal sufficient amount of explanations. The SAG provides an efficient and novel way to avoid overwhelming the user with multiple explanations.

With the user study, the authors sufficiently provide evidence that multiple explanations along with the presented structured visual representation improve the explanatory value of saliency map based XAI methods.


Further remarks:
- Attention maps: When reading the title and abstract it was not clear to me that attention map refers to explanations generated by saliency methods like GradCAM. In defence to the authors, they describe that very prominent in the introduction and the rest of the paper. However, in my opinion, readers could confuse this with an attention-based approach to generate explanations such as [1] and I would recommend changing the wording.

References:

[1] Hiroshi Fukui, Tsubasa Hirakawa, Takayoshi Yamashita, Hironobu Fujiyoshi: Attention Branch Network: Learning of Attention Mechanism for Visual Explanation. CVPR 2019

**Time Spent Reviewing:**

2.5

---

> ### Author Response · Authors · 2021-08-11
> **One Explanation is Not Enough: A Response to cwRt**
>
> Thanks for the reference and your positive review! We will change the wording back to saliency maps in the final version.

---

### Official Review · Reviewer_kZGR · 2021-07-16

**Rating:** 6
**Confidence:** 4

**Summary:**

The paper examines the problem of defining multiple explanations of the decision making of an image classifier. It presents a Structure Attention Graph (SAG) generation approach using a beam search algorithm and diverse sampling, providing visualization of how different combinations of image patches affect the classifier's confidence. It also presents a user study illustrating that users are positively influenced by SAG representations than other methodologies, such as Grad-CAM or I-GOS.

**Ethical Concerns:**

No concerns.

**Limitations And Societal Impact:**

This is OK.

**Main Review:**

It is an interesting paper, tackling the crucial issue of providing better visualization of CNN decision making when dealing with image classification problems. Extending single attention maps to multiple ones can be advantageous in many cases and this is illustrated in the presented study. It first uses beam search to find multiple candidate minimal sufficient explanations, then defining diverse sets of candidates and thereafter building the SAG in a rather straightforward way. The above steps are expressed in terms of error function minimization, but do not include much novelty. The provided study is of value considering the better amount of information provision.
Some specific issues: There is higher computational cost to generate the multiple visualizations outcome.  The paper should provide a measure of this higher computational complexity  when comparing to the other methods' results. Moreover, (and in the supplementary), it should be clearer how the results were obtained (e.g., for the other two methods) or what the percentage in each explanation refers to. Various choices, e.g., of sizes, are made in ad-hoc manner, across the manuscript; how they are made in relation to specific contexts would be needed to formulate.

**Time Spent Reviewing:**

2 hours

---

> ### Author Response · Authors · 2021-08-11
> **One Explanation is Not Enough: A response to kZGR**
>
> Thanks for your review. We will add a comparison of the computational cost and justify the choices we made. For your specific questions:
>
> Computational cost:
>
> The computational cost of beam search is linear in the number of patches tested. The maximal runtime of beam search is 30 seconds (ln. 251-253) but for most images the runtime is significantly less. The combinatorial search runs in exponential time in the number of patches tested hence is not as effective as the beam search.
>
> Results from the baseline methods:
>
> For GradCAM/I-GOS, all the patches were ordered by their weight in the heatmap and we gradually insert the most important patches until the resulting image becomes an MSE (i.e. it can be classified with at least 90% confidence of the full image). Once it becomes an MSE, we consider the minimal explanation by GradCAM/I-GOS to be of that size. The result shows that with beam search we can find significantly more localized explanations than GradCAM/I-GOS for significantly more images.
>
> Choices of sizes:
>
> The size 7x7 was chosen because that’s the maximal size where we can still run a filtered combinatorial search in reasonable time. The m=10 was due to the same reason. The results show that such a filtered combinatorial search is not as effective as beam search. Larger m would make the performance better but also exponentially slower, so we did not attempt that. We will better justify such choices in the final version.

---

> > ### Comment · Reviewer_kZGR · 2021-08-31
> > **My grade remains the same**
> >
> > The authors have provided some responses related to my comments, however they were not really insightful. I still consider that the paper is interesting and can be of value in applications. My grade remains the same.

---

### Decision · Program_Chairs · 2021-09-28

**Decision:**

Accept (Poster)

**Comment:**

The paper revisits (visual) explanations and proposes to utilize a beam search algorithm to systematically search for multiple explanations for each image. To have a compact representation of the explanations, it proposes to make use of structured attention graphs, showing how different combinations of image regions impact the confidence of a classifier. This is an interesting and natural idea and definitely has promise. While the reviews also identify several difficulties such as a representation bias in the experimental evaluation, they also agree that it is important to understand better how to present explanations to the user. And here the paper makes an interesting contributions, as also the most negative reviewer agrees upon.

**Consistency Experiment:**

NeurIPS has a long history of experimentation. In 2014, NeurIPS ran an experiment in which 10% of submissions were reviewed by two independent committees to quantify the randomness in the review process. This year, we repeated a variant of this experiment to see how the quality of the review process has changed over time.  This paper was part of the experiment and was therefore assigned to two committees (consisting of reviewers, an Area Chair, and a Senior Area Chair) that reached independent decisions.  If both committees made the same recommendation, this recommendation was followed. If a single committee recommended acceptance, the paper was accepted (with the exception of a few cases in which the other committee identified what we considered a fatal flaw, e.g., an error in a key result).

This copy’s committee reached the following decision: **Accept (Poster)**

The other committee assigned to the paper recommended **Reject**.  You can find the other set of reviews, along with any follow up discussion with the authors here:
https://openreview.net/forum?id=k5Kbs9uPGP9